# Tide-Surge Interaction observed at Singapore and the east coast of Peninsular Malaysia using a Semi-empirical Model

Zhi Yang Koh[1], Benjamin S. Grandey[1], Dhrubajyoti Samanta[2], Adam D. Switzer[2,3], Benjamin P. Horton[2,3], Justin Dauwels[4], Lock Yue Chew[1]

[1]School of Physical and Mathematical Sciences, Nanyang Technological University, Singapore
[2]Earth Observatory of Singapore, Nanyang Technological University, Singapore
[3]Asian School of the Environment, Nanyang Technological University, Singapore
[4]Department of Microelectronics, Faculty of Electrical Engineering, Mathematics, and Computer Science, Delft University of Technology (TU Delft), The Netherlands

*Correspondence to*: Zhi Yang Koh (kohz0034@e.ntu.edu.sg)

**Abstract.** Tide-surge interaction plays a substantial role in determining the characteristics of coastal water levels over shallow regions. We study the tide-surge interaction observed at seven tide gauges along Singapore and the east coast of Peninsular Malaysia, focusing on the timing of extreme non-tidal residuals relative to tidal high water. We propose a modified statistical framework using a No-Tide-Surge Interaction (No-TSI) null distribution that accounts for asymmetry and variation in the duration of tidal cycles. We find that our modified framework can mitigate false positive signals of tide-surge interaction in this region. We find evidence of tide-surge interaction at all seven locations, with characteristics varying smoothly along the coastline: the highest non-tidal residuals are found to occur most frequently before tidal high water at the south, both before and after tidal high water at the middle, and after tidal high water at the north. We also propose a semi-empirical model to investigate the effects of tidal phase alteration, which is one mechanism of tide-surge interaction. Results of our semi-empirical model reveal that tidal phase alteration caused by storm surges is substantial enough to generate significant change in the timing of extreme non-tidal residuals. To mitigate the effect of tidal phase alteration on return level estimation, skew surge can be used. We conclude that (1) tide-surge interaction influences coastal water levels in this region, (2) our semi-empirical model provides insight into the mechanism of tidal phase alteration, and (3) our No-TSI distribution should be used for similar studies globally.

**Short summary**

Identifying tide-surge interaction (TSI) is a complex task. We enhance existing statistical methods with a more robust test that accounts for complex tides. We also develop a semi-empirical model to investigate the influence of one mechanism of TSI, tidal phase alteration. We apply these techniques to tide-gauge records from Singapore and the east coast of Peninsular Malaysia. We find TSI at all studied locations: tidal phase alteration can change the timing of large surges.

## 1 Introduction

Coastal regions are vulnerable to the combined effects of tides and storm surges, which can induce significant sea-level variations and pose substantial risks to coastal communities and ecosystems (Diaz, 2016; von Storch et al., 2015; Hinkel et al., 2014). Storm surges are rises in sea level brought about by low atmospheric pressure and strong winds acting on the sea surface (Pugh and Woodworth, 2014a). Large storm tides occur when large storm surges coincide with high tides (Stephens et al., 2020; Gregory et al., 2019), which increase the risk of coastal flooding and threaten lives and livelihoods (Pugh and Woodworth, 2014c). The likelihood and impact of such destructive events are further aggravated by sea-level rise (Calafat et al., 2022; Marcos and Woodworth, 2017; Woodworth and Blackman, 2004). To identify the appropriate response to such extreme water level events, we must understand the fundamental processes of tides and storm surges and their mutual interactions.

A dependence between tides and surges has long been noticed at coastal locations (Keers, 1968), and recognised to be caused by interaction between tides and surges (Pugh and Vassie, 1978; Wolf, 1978). Understanding tidal dynamics, surge generation, and their mutual interaction is required to improve operational forecasts of sea levels and the statistical estimation of its extremes (Olbert et al., 2013; Horsburgh and Wilson, 2007; Tawn and Vassie, 1989). This tide-surge interaction is non-linear and can lead to complex coastal dynamics characterized by amplification or attenuation of water levels, which are influenced by local bathymetry, coastline geometry, and atmospheric conditions (Idier et al., 2019). Strong tide-surge interaction has been observed in shallow waters and estuaries (Wolf, 1981) and hence is expected at the Sunda Shelf where Singapore and Malaysia are located. The main mechanism behind

tide-surge interaction is mutual phase alteration (Rossiter, 1961). The generation of surges over a water body is influenced by the depth of the water body. As changes in depth occur partially due to tides, the height of surges can be influenced by tides. The propagation speed of tides is also dependent on depth, which can change due to surges (Proudman, 1957, 1955). Further studies that analysed shallow-water equations along the coast found that the non-linear tide-surge interaction is caused by the advection term, non-linear bottom friction term, and the non-linear shallow water effects of the shallow-water equations (Zhang et al., 2021; Idier et al., 2012).

Studies of tide-surge interaction have employed a range of modelling approaches, including statistical methods (Arns et al., 2020; Haigh et al., 2010; Dixon and Tawn, 1994), numerical models (Costa et al., 2023; Horsburgh and Wilson, 2007; Prandle and Wolf, 1978), and analytical models (Horsburgh and Wilson, 2007). Dixon and Tawn (1994) proposed a statistical framework where they split the tidal range into five equiprobable bands and used a chi-square test to determine whether non-tidal residuals above a height threshold fall uniformly into each band. Horsburgh and Wilson (2007) proposed a modified version of the framework where the tide is instead split into 13 hourly bands between 6.5 hours before and after tidal high water (HW). Horsburgh and Wilson (2007) also provided a simple mathematical explanation for the abundance of large non-tidal residuals at timings halfway between tidal low water and HW.

Application of such frameworks revealed that extreme residuals are most often found 3–5 hours before HW in the Bay of Bengal (Antony and Unnikrishnan, 2013) and the North Sea (Horsburgh and Wilson, 2007), and typically about 2 hours before HW in the English Channel (Haigh et al., 2010). In China and New Zealand, observed tide-surge interaction varies along the coastline: the frequency of extreme residuals peaks before HW at certain locations, after HW at others, and is independent of tides at the remaining locations (Costa et al., 2023; Feng et al., 2019). Numerical models have shown that the inclusion of tide-surge interaction often results in better water level predictions, especially over coastal and shelf waters, whereas the omission of the interaction may lead to under or overestimation of surges at certain locations (Fernández-Montblanc et al., 2019; Idier et al., 2012). For example, Antony et al. (2020) showed that numerically modelled peak water levels generated during Cyclone Aila at the head of the Bay of Bengal would have been overestimated if tide-surge interaction was not simulated.

Here, we focus on investigating tide-surge interaction observed at seven tide gauge locations near Singapore and the east coast of Peninsular Malaysia using modified statistical methods and a new semi-empirical model. Our research objectives are (1) examining the tide gauge records to characterise the tide-surge interaction observed at each tide gauge location and the spatial pattern across locations, and (2) explaining the observed interaction characteristics through a simple semi-empirical model.

We apply a modified version of the statistical method by Horsburgh and Wilson (2007) to determine the presence of tide-surge interaction at each tide gauge location and to characterise these interactions. The existing method groups residuals above a height threshold into 13 hourly bands between 6.5 hours before and after HW, counts the number of residuals in each band and compares the resulting distribution to the uniform distribution using a chi-square test. However, due to asymmetry and variation in the duration of tidal cycles (Guo et al., 2019), the expected null distribution would not be a uniform distribution over 13 hourly bands centred at HW especially at locations with mixed or diurnal tides lasting up to 25 hours. Hence, our key modification to the existing methodology is to replace the uniform distribution with a "No-TSI distribution" as the null distribution. We also use an exact statistical test for hypothesis testing instead of the chi-square test. In addition, we propose a simple approach to compare the tide-surge interaction across locations that have different tidal characteristics (diurnal, mixed, or semidiurnal).

We aim to provide an explicit explanation of the observed tide-surge interaction through our semi-empirical model by combining historical tide and surge data with winds, coastal geometry, and bathymetry. The semi-empirical model accounts for the mechanism of tidal phase alteration: storm surges perturb the depth of the water body which influences the propagation speed of the tide. This results in differences between observed tides and tides predicted from harmonic analysis, which are detected as non-tidal residuals. We use our model to show that this mechanism can significantly influence the timing of extreme residuals to produce signals of tide-surge interaction.

**2 Singapore and the east coast of Peninsular Malaysia**

The seven tide gauges are located within the Sunda Shelf, illustrated in Fig. 1a, which has a typical depth of 40–80 m. The shallowness of this shelf likely enhances the interaction between tide and surge, and its

expanse allows for phase changes in the tides to compound as the tide propagates across the shelf. The eastern coastline of Peninsular Malaysia faces the South China Sea and is exposed to strong seasonal monsoon winds resulting in larger surges (Mohd Anuar et al., 2020). To the north of our study region lies the Gulf of Thailand. To the south of Singapore, the southernmost location of our study region, lies part of the Riau Islands, Sumatra, the Karimata Strait and the Java Sea. The Malacca Strait is west of Singapore and leads to the Indian Ocean. The diurnal and semidiurnal tides that propagate from the South China Sea and the Indian Ocean respectively drive the complex mixed tides in Singapore and the Southern parts of the study region (van Maren and Gerritsen, 2012). At Singapore, the diurnal tides that propagate from the South China Sea get further amplified from reflection against the east coast of Sumatra (van Maren and Gerritsen, 2012).

A close-up of the bathymetry around the seven stations is shown in Fig. S1 and detailed maps of their immediate vicinity are shown in Fig. 1b–h and S2. Of our seven tide gauges, Tanjong Pagar and Johor Baharu are located within straits, Sedili at a river mouth, Tioman between Tioman island and mainland Peninsular Malaysia sheltered from the South China Sea, Kuantan at the mouth of a man-made bay, Cendering on the open coast and Geting sheltered inside a bay.

Monsoonal winds are the main determining factor of hourly surges and their extremes in the Singapore Strait (Tkalich et al., 2013a, 2009). Storm surges occur along the coast throughout the year, reaching 0.35–1.0 m at Geting, the northernmost location of our study region (Abd Razak et al., 2024). The northern part of the coast is more prone to stronger storm surges due to tropical depressions (Mohd Anuar et al., 2023). In the southern part of the coast, a surge of about 1.0 m caused by Typhoon Vamei in 2001 was recorded (Mohd Anuar et al., 2018). Several storm surge events in Singapore reaching 0.3–0.8 m have also been documented since 1974 (Luu et al., 2016), particularly during the winter monsoon period.

Our analysis found that the tidal range in this region can reach 2.7–3.6 m and the largest non-tidal residuals over the past 30 years exceed 0.8 m. Understanding how these components interact and combine provides insight into the contributors to coastal water levels. Hydrodynamical processes, mainly those caused by regional wind forcing during the seasonal monsoons, have shown a strong influence on the

water levels at the seven tide gauges analysed in this study (Tay et al., 2016; Luu et al., 2016; Kurniawan
et al., 2015; Karri et al., 2014; Tkalich et al., 2013a; Chen et al., 2012).

## 3 Data and methods

The general methodology begins with separating hourly tide gauge data into tides and non-tidal residuals.
Following the methodology of Horsburgh and Wilson (2007), we identify the largest non-tidal residuals
and find their timings relative to their nearest tidal high water and tabulate it as a frequency distribution.
Next, we calculate the No-TSI distribution. We compare the frequency distribution to the No-TSI
distribution using hypothesis testing to determine the presence of tide-surge interaction. We then generate
model residuals using our semi-empirical model and repeat the above procedure to determine if tide-surge
interactions are generated by our model. We use research quality hourly tide gauge data obtained from
the University of Hawaii Sea Level Center. Bathymetry, hourly 10 m winds and mean sea level pressure
are obtained from ERA5.

### 3.1 Tide gauge data, tides and residuals

Research quality hourly water level from tide gauges at seven locations (Fig. 1) along Singapore and the
east coast of Peninsular Malaysia are obtained from the University of Hawaii Sea Level Center (Caldwell
et al., 2001). Details of the tide gauge records are tabulated in Table 1. Observations have been made over
at least 29 years at each location with a data completion rate of 95–99 %. The length of these records is
close to the 30-year threshold typically considered the minimum length required for analysis of extreme
sea levels (Rasmussen et al., 2018). Nonetheless, these stations have some of the longest records within
Southeast Asia (Caldwell et al., 2001).
To compute the tidal level and non-tidal residuals, we use the equation
$$X(t) = Z(t) + T(t) + R(t). \tag{1}$$
$X(t)$ is the hourly water level recorded by a tide gauge at time $t$. $Z(t)$ is the 1-year (8766-hour after
accounting for leap years) moving average of $X(t)$. A 1-year window was chosen because intra-annual

behaviour of residuals is well understood to be periodic due to seasonal variations, and its influence on residuals is of interest in this study (Tkalich et al., 2013b). Detrending the hourly water level by the 1-year moving average removes trends of annual and longer timescales from the tide gauge records. The 1-year moving average is calculated only if at least 85 % of the data in the 8766-hour window is available. This reduces our usable data to 87–99 % across the seven locations. The standard deviation and certain key quantiles of the detrended water level, $X(t) - Z(t)$, are tabulated in Table 2. While water levels of this height do not pose an immediate threat to the region, understanding the underlying drivers is integral for applications such as early detection of extreme events, forecasting, or coastal protection design purposes especially with sea level rise and the increasing frequency and intensity of storms (Mohd Anuar et al., 2018). $T(t)$ is the tidal level which we estimate using UTide, a tidal harmonic analysis package implemented in Matlab (Codiga, 2011). $X(t) - Z(t)$ is split into two halves based on the start and end dates in Table 1 and used as inputs to UTide, as UTide does not recommend using records longer than 18.6 years as input. UTide was used to identify the amplitudes and phases of 66 tidal constituents with periods of up to 32 days through harmonic analysis. Constituents with a signal-to-noise ratio of at least 2 are used to construct $T(t)$. $R(t)$, the residual, is then estimated as $R(t) = X(t) - Z(t) - T(t)$. We denote the residual obtained through this procedure as $R_{\text{gauge}}$. The time-series of $X(t) - Z(t), T(t)$ and $R_{\text{gauge}}$ at each tide gauge during their largest recorded $R_{\text{gauge}}$ are illustrated in Fig. S3.

A summary of the tidal characteristics at the seven locations is tabulated in Table 3. The tides at these locations lie within the microtidal (<2 m) and mesotidal (2–4 m) ranges with diurnal tidal ranges of close to 2 m except at Cendering and Geting where diurnal tidal ranges are 1.5 m and 0.8 m respectively. We found that the daily tidal range can be as large as 3.6 m at Johor Baharu and Kuantan, similar to other studies (Ismail et al., 2018; Ng and Sivasothi). In a shelf with a depth of 40–80 m, tides of 3.6 m produce a water depth deviation from its mean by up to about ±4 %. The tides at a location can be categorised into diurnal, semi-diurnal, or mixed using the tidal form factor $F$ which compares the relative importance of the following diurnal and semidiurnal tidal constituents (Pugh and Woodworth, 2014b):

$$F = \frac{K_1 + O_1}{M_2 + S_2}. \tag{2}$$


A common classification considers a location with a factor of <0.25 to be semidiurnal, 0.25–1.50 to be
mixed with mainly semidiurnal tides, 1.50–3.00 to be mixed with mainly diurnal tides and >3.00 to be
diurnal (Pugh and Woodworth, 2014b). Based on this classification, the tides at all stations are identified
to be mixed, with Cendering and Geting being mainly diurnal while all other stations to the south are
mainly semidiurnal. Out of all tidal constituents estimated across seven locations, we find that the
amplitude of most constituents does not change by more than 5 mm between the first and second half of
their tide gauge records while eight constituents saw a change of 5–13 mm. The change in mean diurnal
tidal range is about 0.2 cm at Tanjong Pagar to 4.9 cm at Geting. Attribution of these changes remains a
difficult task and is outside the scope of this study, but some local drivers include (1) dissipation and
turbulent mixing; (2) depth of channels and flats; (3) surface area, width, and convergence; (4) resonance
and reflection; (5) river flow; and (6) changes in instrumentation (Haigh et al., 2020).
**3.2 Identifying tide-surge interaction**
To determine whether tide-surge interaction is present at each of the seven tide gauge locations, we apply
a modified version of the statistical method by Horsburgh and Wilson (2007) on the processed tide gauge
records (Sect. 3.1). This method compares the timing of extreme residuals to the nearest HW.
To identify extreme residuals, the 99th percentile and above of $R_{gauge}$ are identified. There are
many clusters of $R_{gauge}$ above this threshold, where a cluster refers to a collection of $R_{gauge}$
consecutively measured, each measured 1 hour from the last. The largest $R_{gauge}$ in each cluster is
identified and sorted. Starting from the largest, the sorted $R_{gauge}$ are added to the set of extreme $R_{gauge}$
unless the $R_{gauge}$ was measured within 1 week (168 hours) of another extreme $R_{gauge}$ in the set. Some
other thresholds used in published studies range from 12–60 hours (Arns et al., 2020; Feng et al., 2019;
Rasmussen et al., 2018; Buchanan et al., 2016; Horsburgh and Wilson, 2007). We choose a threshold of
1 week to reduce the odds of double counting long-lasting surges, as such events can last between 1 and
5 days in this region (Meteorological Service Singapore; Cannaby et al., 2016; Marzin et al., 2015;
Tkalich et al., 2009). We also found statistically significant autocorrelation in the $R_{gauge}$ time-series at
time lags shorter than 168 hours. A declustering threshold of 168 hours still retain sufficient observations
for our analysis.
To compare locations with predominantly semidiurnal tides with locations with predominantly
diurnal tides, we split the extreme $R_{\text{gauge}}$ into two groups. We define a tidal cycle as the duration from
one local minima in $T(t)$ to the observation immediately preceding the next local minima as illustrated
in Fig. 2b. One group of extreme $R_{\text{gauge}}$ were observed during tidal cycles of 21 hours or shorter,
representing extreme residuals that occurred during semidiurnal cycles. The other group contain extreme
$R_{\text{gauge}}$ observed during tidal cycles of at least 22 hours, representing extreme residuals that had occurred
during diurnal tides. This separation between 21-hours-or-less and 22-hours-or-more was chosen based
on the characteristics of the duration of tidal cycles at the seven locations, where the distribution of the
duration of tidal cycles was found to be bimodal at each location and the 22-hour mark tends to distinguish
the two modes (Fig. S4).
The HW in the same tidal cycle as each extreme $R_{\text{gauge}}$ is identified and the timing difference
between the extreme $R_{\text{gauge}}$ and the respective HW is quantified at hourly resolution. Across the set of
extreme $R_{\text{gauge}}$, the frequency of extreme $R_{\text{gauge}}$ found a certain number of hours relative to HW, $h$, is
counted. This frequency is plotted as a *frequency distribution* in the form of a histogram (Fig. 3). The
magnitude of each extreme $R_{\text{gauge}}$ is represented by colour in the frequency distribution. Box plots below
the histograms show the median and its 95 % confidence interval, the interquartile range (IQR), a range
that extends up to 1.5×IQR from the limits of the IQR, and the outliers (Fig. 3).
We use the frequency distribution to test the null hypothesis that assumes that there is no tide-
surge interaction. To do so, we test whether the frequency distribution is drawn from a null distribution
representing a scenario where extreme events are equally likely to occur at any stage of a tidal cycle.

**3.3 No-Tide-Surge Interaction distribution**

In existing studies, the null distribution is a uniform distribution with a value of $n/13$ at $h =$
$-6, -5, \dots, 0, \dots, 5, 6$ where $n$ is the total number of extreme events (Horsburgh and Wilson, 2007; Haigh
et al., 2010; Antony and Unnikrishnan, 2013; Feng et al., 2019; Costa et al., 2023). It assumes that tidal
cycles are always 13 hours long (i.e. semi-diurnal) and HW is always at the mid-point of this cycle (i.e.

tides are always symmetric). Hence, if there are $n$ extreme events, we expect $n/13$ events to happen at each hourly band if extreme events are equally likely to occur at any stage of a tidal cycle. However, as tides are not always semi-diurnal and symmetric, the uniform distribution is not the most suitable null distribution to represent the null hypothesis. Instead, we propose a "No-Tide-Surge Interaction distribution" or "No-TSI distribution" as the null distribution. The No-TSI distribution is the *expected* frequency distribution in the absence of tide-surge interactions. Figure 2 illustrates how this distribution is obtained. The No-TSI distribution is empirically derived from $T(t)$, the tidal level obtained by applying UTide to the tide gauge records at each location. It is a distribution that depends on the local tidal characteristics and hence is location specific. For a given location, the null distribution is generally non-uniform because the length of the tidal cycle varies. For example, tidal cycles of 14 hours or longer are relatively rare at Tanjong Pagar, so randomly selected times that occur at 7 hours from the nearest tidal HW will also be relatively rare (Fig. 3a). This leads to non-uniform sampling of the number of hours from the nearest tidal HW. The No-TSI distribution corresponds to uniform sampling in time, which is non-uniform with respect to the number of hours from the nearest tidal HW. Thus, the No-TSI distribution is obtained by counting the number of tide-gauge observations found at a certain number of hours relative to HW as illustrated in Fig. 2.

The No-TSI distribution allows us to account for the complex mixed tides at each location, which causes the tidal cycles at the coast to have a period of any duration from 8 to 26 hours (Fig. S4). Let $f(h)$ be the number of tide gauge measurements collected at $h$ hours from HW. Assuming independence between tide and surge, the probability that an extreme event will be found at $h$ is $p_h = f(h)/\sum_h f(h)$: the normalized frequency of $f(h)$. Note that $p_h$ is a probability mass function over the domain $h$: a function that tells us the probability of an extreme event happening at $h$ is $p_h$. Letting $n$ be the number of extreme events that occurred at this tide gauge over its length of records, the No-TSI distribution is simply $n \cdot p_h$.

We can also find the 95% confidence interval of the No-TSI distribution. Assuming that extreme events are mutually independent, the probability that $k_h$ out of $n$ extremes are found at $h$ hours from HW follows the binomial distribution $k_h \sim \text{Bin}(n, p_h)$. We compute the 2.5th and 97.5th percentiles of the binomial distribution at each $h$ to obtain the 95% confidence interval of the No-TSI distribution.

For testing the null hypothesis, a bootstrapping method is used to calculate $p$-values (Appendix
A) with 1,000,000 bootstrap samples. To account for family-wise error rate due to our multiple
comparisons at seven locations, we apply the Bonferroni correction to our chosen significance level of
0.05 and require a $p$-value below $0.05/7 = 0.007$ and $0.05/4 = 0.0125$ to reject the null hypothesis
when testing for tide-surge interaction during semi-diurnal and diurnal cycles respectively. This method
is chosen over the usual approach of the chi-square test due to our relatively small sample size $n$. As the
chi-square test approximates the binomial distribution of $k_h$ with a normal distribution, a common rule
of thumb requires $n \cdot p_h \geq 5$ for a decent approximation. This criterion is not satisfied for most values of
$h$ at all seven locations.

## 3.4 Semi-empirical model

One effect of tide-surge interaction is tidal phase alteration, where surge-caused increase in water depth
advances the timing of HW. Our semi-empirical model aims to investigate the first-order effects of tidal
phase alteration on the height and timing of residuals due to storm surges driven by wind and the inverse
barometer effect. To do so, we first consider the regional context. Singapore and the east coast of
Peninsular Malaysia are located within the Sunda Shelf (Fig. 1). The typical bathymetry of 40–80 m in
this region of the Sunda shelf terminates at about 700 km away from the coast, declining steeply at the
edge of the Sunda Shelf where it borders the South China Sea, which has depths of 4,000 m and deeper.
It is within the shallow region that tide-surge interaction is the strongest. Hence, for our semi-empirical
model, we estimate the effects of tidal phase alteration due to surges generated within the ocean bounded
by the red rectangle in Fig. 1. A rectangular region was chosen for simplicity, with one edge passing
through the Tanjong Pagar and Geting gauge locations and the remaining three edges encompassing as
much of the shallow shelf as possible. The edges of the rectangle parallel to the Malaysian coast are
roughly 833 km long and are separated by a longitude of 6.5°, making the edges perpendicular to the
coast roughly 759 km long. To compare the storm surge at the seven tide gauge locations of interest, we
assume that the surges at these locations result from forcings over the same region. We model surges as
$$R_{\text{sum}} = R_{\text{baro}} + R_{\text{wind}} + R_{\text{phase}}, \qquad\qquad (3)$$
where $R_{\text{baro}}$ is the water level response to the inverse barometer effect, $R_{\text{wind}}$ is the wind-driven surge
and $R_{\text{phase}}$ is the contribution by tidal phase alteration. $R_{\text{sum}}$ corresponds to a model estimate of $R_{\text{gauge}}$.

To obtain $R_{\text{baro}}$, we assume hydrostatic balance and use the inverse barometer effect by using a

25-hour moving average of pressure:

$$R_{\text{baro}} = -\frac{\overline{\langle\Delta P\rangle_A}}{\rho g}, \tag{4}$$

where $\Delta P$ is the deviation of sea-level pressure from the global mean, $\langle\Delta P\rangle_A$ represents a spatial average
of $\Delta P$ over the ocean within the red rectangle in Fig. 1 and $\overline{\langle\Delta P\rangle_A}$ represents a 25-hour moving average
of $\langle\Delta P\rangle_A$. $\rho$ is the density of seawater, $g$ is the gravitational acceleration and $1/\rho g = 9.9 \times 10^{-5}$ m Pa$^{-1}$
(Gregory et al., 2019, Pugh and Woodworth, 2014c).

The unique topography of this region with an extensive area that is shallow and relatively uniform

in depth has led us to adopt a simplified version of the sea-level gradient equation to estimate $R_{\text{wind}}$ using
wind velocity:

$$\frac{\partial \zeta}{\partial x} = \frac{\rho_{\text{air}} C_d |W| W \cdot \hat{x}}{\rho g D}, \tag{5}$$

where we have assumed that the coastal sea is shallow enough to keep only terms with water depth in the
denominator but is deep enough to justify ignoring bottom stress (Pugh and Woodworth, 2014c). In Eq.
(5), $\zeta$ is the sea level and $x$ is spatial displacement in a specified direction, making $\partial\zeta/\partial x$ the sea-level
gradient along $x$. $\rho_{\text{air}}$ is the density of air, $C_d$ is the drag coefficient at the sea surface, $\rho$ is the density of
seawater, $g$ is the gravitational acceleration, $D$ is the undisturbed water depth, $W$ is the wind velocity
vector, and $\hat{x}$ is a unit vector parallel to $x$. $x$ is defined to be perpendicular to the edge across Tanjong
Pagar and Geting in Fig. 1. In this region, $g = 9.78$ ms$^{-2}$. Bathymetry influences the height of surges
through the term $D$ in the denominator of Eq. (5), causing shallow regions to experience larger surges
when subject to the same wind forcing.
We further assume that $\rho_{\text{air}}$, $C_d$, $\rho$ and $g$ are spatially and temporally homogeneous over the
bounded region (Gregory et al., 2019) and that $C_d$ is independent of wind speed (Wróbel-Niedzwiecka et
al., 2019). Surges are not only a product of instantaneous winds but are also partially a result of winds
over a past number of hours. To account for the time taken for the winds over the bounded region to cause
surges at the tide gauge locations, we use a 25-hour running average of Eq. (5) to estimate the wind-driven
surge. Based on those assumptions, we estimate $R_{\text{wind}}$ by discretising Eq. (5) and numerically integrating
over the 759 km along $\hat{x}$:

$$R_{\text{wind}} = \frac{\rho_{\text{air}} C_d}{\rho g} \int \overline{\frac{|W|W \cdot \hat{x}}{D}} dx = k \, \overline{\langle \frac{|W|W \cdot \hat{x}}{D} \rangle_A}, \qquad (6)$$

where $W$ is the wind velocity with its zonal ($u$) and meridional ($v$) components obtained from the hourly
10m winds of ERA5 (Hersbach et al., 2018), $\hat{x}$ is a unit vector along $x$ pointing towards the Malaysian
east coast as shown in Fig. 1, $\langle |W|W \cdot \hat{x}/D \rangle_A$ represents a spatial average of $|W|W \cdot \hat{x}/D$ over the ocean
within the red rectangle in Fig. 1, $\overline{\langle |W|W \cdot \hat{x}/D \rangle_A}$ represents a 25-hour moving average of
$\langle |W|W \cdot \hat{x}/D \rangle_A$, $k = \rho_{\text{air}} C_d L_{\text{wind}}/\rho g$ and $L_{\text{wind}} = 759$ km is the distance which Eq. (5) was integrated
over. While most of the constants in $k$ are known, $C_d$ is often used as a final tuning parameter in non-
tidal barotropic models (Zweers et al., 2012, Kurniawan et al., 2015). Using Eq. (3), Eq. (6) and
considering that $|R_{\text{phase}}| \ll |R_{\text{wind}}|$, we fit $\overline{\langle |W|W \cdot \hat{x}/D \rangle_A}$ to $R_{\text{gauge}} - R_{\text{baro}}$. We use a simple linear
regression to obtain the constant $k$ and our estimate for $R_{\text{wind}}$ at each location.
We obtain our model storm surge as

$$R_{\text{surge}} = R_{\text{wind}} + R_{\text{baro}} = k \, \overline{\langle \frac{|W|W \cdot \hat{x}}{D} \rangle_A} - \frac{\overline{\langle \Delta P \rangle_A}}{\rho g}. \qquad (7)$$

To estimate $R_{\text{phase}}$, we consider the influence of depth on the propagation speed of tidal waves.
With the speed of tidal waves given by $c = \sqrt{gD}$ (Pugh and Woodworth, 2014c) and treating $R_{\text{surge}}$ as a
perturbation to the undisturbed water depth $D$, the tide advancement time caused by a change in $D$ due to
$R_{surge}$ is

$$\Delta t = \frac{L_{tide}}{\sqrt{gD}} - \frac{L_{tide}}{\sqrt{g(D+R_{baro}+R_{wind}/2)}},$$        (8)

where $L_{tide}$ is the distance travelled by the tidal wave along the shelf. This considers that $R_{baro}$ is the
water level response to $\Delta P$ averaged over the ocean within the red rectangle in Fig. 1 while $R_{wind}$ is
obtained by integrating the sea level gradient over the same area. Hence, we approximate the average
depth perturbation due to $R_{wind}$ over this area as $R_{wind}/2$. We also assume that the tides travel straight
towards the coast in the direction of $\hat{x}$, allowing us to equate $L_{tide} = L_{wind}$. We then calculate the effects
on residual height due to tide advancement from $R_{surge}$ as $R_{phase} = T(t + \Delta t) - T(t)$. $R_{phase}$ can be
viewed as a phase shift in the tidal levels, where extreme $R_{phase}$ tend to cluster on the rising or falling
tides instead of during tidal high or low water (Horsburgh and Wilson, 2007).

Finally, we use the procedure described in Sect. 3.2 to obtain the timing of extreme $R_{surge}$, $R_{phase}$

and $R_{sum}$ relative to HW and plot their frequency distributions. To determine their contributions to tide-
surge interaction, the bootstrapping method is used to identify the presence of tide-surge interaction in
$R_{surge}$ , $R_{phase}$ and $R_{sum}$ . The frequency distributions of $R_{sum}$ are compared to the frequency
distributions of $R_{gauge}$ to evaluate whether $R_{sum}$ can reproduce the tide-surge interaction found in the
tide-gauge records.
**4 Results and discussion**
**4.1 Observed tide-surge interaction**
The frequency distributions of $R_{gauge}$ at all seven locations are shown in Fig. 3, all of which deviate
significantly from their respective No-TSI distribution and provide evidence of tide-surge interaction at
every location. Based on how we have defined extremes, we find that extreme $R_{gauge}$ range from 362 mm
at Tanjong Pagar (Fig. 3a) to 1195 mm at Geting (Fig. 3g), with the smallest extremes occurring at the
eastern stations and increasing as we travel north. This is expected as the northern part of the coast is

more prone to larger storm surges due to tropical depressions (Mohd Anuar et al., 2023). The extremes, especially the largest extremes at each location, are unlikely to happen close to HW at all seven locations. This means that while the $R_{gauge}$ exceeding the 99th percentile can occur close to HW, the peak of each cluster of exceedances is unlikely to be found in the time window close to HW. This also means that the largest residuals are unlikely to coincide with high tides to form large storm tides. Across the locations, this time window generally begins 2–3 hours before HW and ends 3–5 hours after HW. Beyond this time window, the frequency of extreme $R_{gauge}$ increases, giving us frequency distributions that are mostly bimodal. At the four southernmost stations, we find the primary mode of their frequency distributions before HW: at -5 hours at Tanjong Pagar (Fig. 3a) and Sedili (Fig. 3c), at -6 hours at Johor Baharu (Fig. 3b) and at -7 hours at Tioman (Fig. 3d). Outside Southeast Asia, similar signals have been found at Port Otago in New Zealand (Costa et al., 2023), Shijiusuo, Lianyungang and Xiamen along the coast of China (Feng et al., 2019), Hiron Point at the Bay of Bengal India (Antony and Unnikrishnan, 2013), and Aberdeen, North Shields, Immingham, Cromer and Sheerness at the North Sea (Horsburgh and Wilson, 2007). At Kuantan, Cendering and Geting (Fig. 3e–3g), the primary mode is found after HW, at +6, +6 and +4 hours respectively. Outside Southeast Asia, such signals have appeared less commonly but have been found at Onehunga in New Zealand (Costa et al., 2023) and at Kaohsiung and Zhapo in China (Feng et al., 2019).

Comparing our results between the seven tide gauges along Singapore and the east coast of Peninsular Malaysia, we find a spatial pattern in the tide-surge interaction. At the southernmost stations of Tanjong Pagar, Johor Baharu and Sedili (Fig. 3a–c), the mass of the frequency distribution is heavily concentrated around their primary modes, which are found before HW. At Tioman (Fig. 3d), the primary mode is still found before HW, but the secondary mode has a distinctly heavier weight than the previous three stations. At Kuantan (Fig. 3e), tide-surge interaction has crossed over to another regime where the primary mode occurs after HW, but the secondary mode found before HW still carries comparative weight. At Cendering and Geting (Fig. 3f–g), the northernmost stations, the primary mode after HW is much heavier than the secondary mode before HW. This spatial pattern can also be seen using the box plots, which are compiled in Fig. 4a. The tight interquartile range at Tanjong Pagar, Johor Baharu and Sedili shows that the mass of their frequency distributions is concentrated at -6 to -4 hours relative to

HW. The larger interquartile range at Tioman shows that there is a more equal mass between the two modes, with the median at -5 hour and mode at -7 hour revealing that the distribution is still heavier towards the negative values. The opposite is true at Kuantan, with its similar interquartile range to Tioman but with its median at -0.5 and mode at +6 instead. At Cendering and Geting, the lower quartile is closer to HW, showing that the difference in relative mass between the two modes has increased.

A transition in the tide-surge interaction characteristic, based on a significant (95% confidence interval) change in the medians from negative to positive values in Fig.4a, happens between Tioman and Geting which are located about 440 km apart. This is a relatively short distance compared to studies that have seen such transition such as 700 km between Zhapo and Xiamen in China (Feng et al., 2019) and 1,650 km between Otago and Onehunga in New Zealand (Costa et al., 2023). Feng et al. (2019) suggested that this transition may be related to the ratio between the amplitude of tidal constituents $M_2$ and $K_1$, where extremes tend towards before HW at locations with larger $M_2$ and after HW when $M_2$ is small. We observe the same transition in our study region, where $M_2/K_1 < 1$ at the northernmost stations of Cendering and Geting, $M_2/K_1 = 1$ at Kuantan and $M_2/K_1 > 1$ at the remaining four stations to the south (Table 3).

Using the quantitative test (hypothesis testing against the No-TSI distribution), we verify that the $p$-value of obtaining the frequency distribution from the No-TSI distribution is below the required level at all seven locations, allowing us to reject the null hypothesis that the frequency distribution was drawn from the No-TSI distribution at a significance level of 0.05 (Fig. S5). This provides further evidence supporting the presence of tide-surge interaction during semidiurnal tides at all seven tide gauge locations studied. The presence of tide-surge interaction at these gauges was expected given the shallow regional bathymetry and their proximity of tide-gauges to narrow waterways connecting the Pacific, Indian Ocean, and Java Sea. Locations like the North Sea, English Channel, and some parts along the coast of China where tide-surge interactions have been observed also have similar geographical properties.

During diurnal tides, we found no extreme $R_{gauge}$ at Tanjong Pagar, Johor Baharu and Sedili. Few extreme $R_{gauge}$ were found during diurnal tidal cycles at Tioman and Kuantan (Fig. S6a–b), while many were found at Cendering and Geting (Fig. S6c–d). This is expected as Cendering and Geting experience mainly diurnal tides, in contrast to the other locations. The lack of observations at the five other stations

is due to the rare occurrence of diurnal tides at these locations, which is shown in Fig. S4. The extreme
$R_{\text{gauge}}$ during diurnal tides shares the same spatial characteristics as the extremes during semidiurnal
tides, starting from 410 mm in the South at Tioman (Fig. S6d) and increasing up north to 992 mm at
Geting (Fig. S6g). No extremes are found within two hours of HW. Two observations are available at
Tioman, which are too few to confidently determine the presence of tide-surge interaction even though
both observations were found at least 6 hours after HW. The same can be said for Kuantan, where seven
observations are available and were mostly found at least 6 hours after HW. Nonetheless, we calculate
their $p$-values and compare them to $0.05/4 = 0.0125$. We find that their $p$-values are insufficient to
reject the null hypothesis that the observed frequency distribution was drawn from the No-TSI distribution
at a significance level of 0.05 and fails to provide evidence of tide-surge interaction during diurnal tidal
cycles at these two locations. The results at these two locations are more likely due to a lack of sufficient
data than an indication of the absence of any tide-surge interaction, and we expect to see tide-surge
interaction based on the properties of the geography discussed earlier. At Cendering and Geting, we
continue to see the pattern where extreme $R_{\text{gauge}}$ are unlikely to happen close to HW. This leads to a
bimodal distribution with the primary modes at both locations found after HW, like their semidiurnal
counterparts. The mode is 14 hours after HW at Cendering and 13 hours after HW at Geting. Respective
$p$-values provide evidence of tide-surge interaction at both locations (Fig. S7) as expected at these
locations.

## 4.2 Semi-empirical model results

We obtain $R_{\text{surge}}$ by fitting $\overline{\langle |\boldsymbol{W}|\boldsymbol{W} \cdot \hat{\boldsymbol{x}}/D\rangle_A}$ to $R_{\text{gauge}} - R_{\text{baro}}$ as described in Sect. 3.4. $R_{\text{surge}}$ has a
correlation of 0.7–0.8 with the tide gauge residuals (Fig. S8). This corresponds to an explained variance
(coefficient of determination) of 0.5–0.6. This suggests $R_{\text{surge}}$ is an adequate proxy of storm surges.
We obtain the timing of extreme $R_{\text{surge}}$ as a frequency distribution using the procedure described
in Sect. 3.2 and compare it to its No-TSI distribution using hypothesis testing to determine whether there
is any signal of tide-surge interaction in $R_{\text{surge}}$. The reason for doing so is to show that the observed tide-
surge interaction between $R_{\text{gauge}}$ and tide is not caused by any correlation between wind and tide or wind

and atmospheric sea level pressure, which are generated by independent processes. The validation of this assumption would imply that the observed dependence between $R_{\text{gauge}}$ and tide is not caused by possible correlation to a common third independent variable, but that tide-surge interaction is indeed present. We find that the resulting frequency distributions for $R_{\text{surge}}$ do not deviate significantly from their No-TSI distribution and provide no evidence of dependence between $R_{\text{surge}}$ and tide (Fig. S9–S12), indicating the absence of such a confounding variable.

To estimate the influence of tidal phase modulation, we compute $\Delta t$ using Eq. (8) and then calculate $R_{\text{phase}} = T(t + \Delta t) - T(t)$ (Sect. 3.4). We apply the procedure of Sect. 3.2 to obtain the extremes of $R_{\text{phase}}$ and find a clear dependence between $R_{\text{phase}}$ and tide (Fig. S13–S16). During semi-diurnal tidal cycles, extreme values of $R_{\text{phase}}$ are mostly found 2–4 hours before HW at all seven locations (Fig. S13). During diurnal cycles, extreme values are mostly found 3–5 hours before HW (Fig. S15). As with $R_{\text{gauge}}$ and $R_{\text{wind}}$, we found no extremes during diurnal cycles at Tanjong Pagar, Johor Baharu and Sedili.

The prevalence of extreme $R_{\text{phase}}$ within a narrow window of time relative to HW is due to $R_{\text{phase}}$ being largest at one-fourth of a tidal cycle before HW, as illustrated in Fig. S17. As the natural period of a semi-diurnal tidal cycle is about 12–13 hours, a sinusoidal tidal waveform has the steepest gradients about 3 hours from its local maxima. This results in the tidal waveform having the greatest difference from a slightly horizontally displaced copy at close to 3 hours from HW. Horsburgh and Wilson (2007) describe this mechanism in detail. However, extreme values of $R_{\text{phase}}$ are not found at 6 hours before HW during diurnal tides. This is because a sinusoidal wave with amplitude $A$ and frequency $\omega$ has a gradient that is proportional to the product $A\omega$. Since semidiurnal components of tides have about twice the frequency of their diurnal counterparts, diurnal constituents need to have at least twice the amplitude of semidiurnal constituents to have the same or stronger influence on $R_{\text{phase}}$. The tidal form factor $F$, which computes the ratio between the amplitudes of the main diurnal and semidiurnal constituents, can provide a measure of this influence. While the tides at Cendering and Geting are mainly diurnal, the tidal form factors at both locations are lower than 2 which indicates that the diurnal constituents have

amplitudes less than twice that of the semidiurnal constituents (Table 3). Therefore, the semidiurnal signal has a stronger influence on the timing of extreme $R_{\text{phase}}$.

The frequency distributions of $R_{\text{sum}}$ at the seven locations during semidiurnal tidal cycles are shown in Fig. 5. The frequency distributions and $p$-values for $R_{\text{sum}}$ suggest the presence of tide-surge interaction at Tanjong Pagar, Johor Baharu, Sedili, Tioman, Kuantan and Cendering (Fig. 5a–f and S18a–f). No significant interaction is identified at Geting (Fig. 5g and S15g). This can also be seen in Fig. 4b where the interquartile range of the frequency distributions at Geting do not deviate from zero while the interquartile range at the other six locations do. Figure 5 reveals that our semi-empirical model predicts frequency distribution of $R_{\text{sum}}$ at all seven locations to have a single mode as opposed to the bimodal frequency distribution of the tide gauge residuals. At the six locations where $R_{\text{sum}}$ provides an indication of tide-surge interaction, the modes of their frequency distributions of $R_{\text{sum}}$ lie within 2–4 hours before HW, following $R_{\text{phase}}$. During diurnal tidal cycles, we find evidence of tide-surge interaction only at Cendering (Fig. S19c and 20c) while no evidence of tide-surge interactions can be seen at Tioman, Kuantan and Geting (Fig. S19a, b, d, S20a, b, d). As with their $R_{\text{gauge}}$ counterparts, the hypothesis testing results at Tioman and Kuantan are likely due to a lack of sufficient observations (Fig. S19a–b). The result at Geting, which is negative for tide-surge interactions, is likely due to its relatively weaker diurnal constituents. This can be seen from the tidal form factor $F$ in Table 3, where the form factor of 1.56 at Geting is lower than 1.90 at Cendering which tested positive for tide-surge interactions.

We find that $R_{\text{phase}}$ can significantly influence the distribution of the extreme values of $R_{\text{sum}}$, indicating that the process of tidal phase alteration—where surges perturb the depth of the water body and influences the propagation speed of the tide—produces a significant and measurable tide-surge interaction at six locations. This is despite $R_{\text{phase}}$ contributing to <1% of the variance of $R_{\text{sum}}$ at all seven tide gauge locations (Fig. S21). By an alternative metric, the ratio between the standard deviation of $R_{\text{phase}}$ and the standard deviation of $R_{\text{surge}}$ is only 0–2%. Thus, while the magnitude of $R_{\text{sum}}$ is effectively fully dependent on $R_{\text{surge}}$, the timing of its largest values is dependent on $R_{\text{phase}}$, indicating the significant contribution of tidal phase alteration to tide-surge interaction. Our findings complement those of Chen et al. (2012), who found that the influence of tide-surge interactions on large surges is

negligible, although tide-surge interaction may alter the time of tidal high and low water. Accounting for tide-surge interaction would be important in applications such as forecasting where timing is important and has been found to improve water level predictions in this region and other parts of the world (Antony et al., 2020; Fernández-Montblanc et al., 2019; Kurniawan et al., 2015; Idier et al., 2012).

Comparing Fig 4b to 4a, our semi-empirical model (Fig. 4b and 5) is unable to accurately predict all the characteristics of tide-surge interaction found in $R_{gauge}$ (Fig. 3 and 4a). At Tanjong Pagar, Johor Baharu, Sedili and Tioman, extreme residuals typically occur even earlier than what our model suggests. Tidal phase alteration shifts the timings of extreme residuals towards times where tides are rising or falling the quickest, and no further. Our model is also unable to produce the bimodal distribution found in the tide gauge data illustrated in Fig. 3 and fails to produce statistically significant tide-surge interaction at Geting. This suggests that the mechanism of tidal phase alteration cannot fully explain the observed tide-surge interaction and that additional explanations are required to fully account for the observations.

One possible contributor is that shallower water during low tides can result in higher surges from near-shore winds (Pugh and Woodworth, 2014c). Changes in tides due to tidal phase alteration can affect the height of surges through Eq. (5), while surges further contribute to tidal phase alteration through Eq. (8) causing the mutually interacting process of mutual phase alteration between tides and surges. Kurniawan et al. (2014) found that the tidal cycle does influence non-tidal residuals in this region. This could result in a further shift of the model extremes towards tidal low water, which would be closer to the observed tide-surge interaction. The impact of non-linear effects resulting from bottom friction and momentum advection also cannot be overlooked in coastal areas and are significant in this region (Kurniawan et al., 2015). We were not able to consider the effects of water flow between the Pacific and Indian Ocean through the Singapore and Johor Strait on the results at Tanjong Pagar and Johor Baharu. We also could not account for the tide-surge-river interactions at Sedili, the presence of Tioman island at Tioman, and how the Gulf of Thailand affects tide-surge interactions at the northern tide gauges. Our semi-empirical model can reveal macroscopic properties of tide-surge interactions but is unable to reveal detailed spatial and temporal properties at high resolution like alternative numerical models. To reveal such details, the use of dynamical models or partial differential equation solvers would be needed. For instance, Kurniawan et al. (2015) and Tay et al. (2016) used a multi-scale depth-integrated hydrodynamic

model that simulated hydrodynamics at a higher spatial and temporal resolution and found significant contribution in the order of 10 cm to non-tidal residuals by tide-surge interactions. However, such numerical models are complex, difficult to tune, and computationally intensive. While it does not replace hydrodynamical models, our semi-empirical model offers theoretical insights by providing a clear and transparent explanation of tidal phase alteration and its impact at a low computational cost.

The semi-empirical model is less applicable elsewhere because it has been designed for the study region. However, the semi-empirical model could be modified for other coastlines situated along a shallow shelf. For instance, the physical characteristic of our study region—a semi-enclosed shallow region leading out to steep depths—can be found on a smaller spatial scale at the Yellow Sea and North Sea and our semi-empirical model may find similar success there.

Testing our No-TSI distribution against the uniform distribution, we found false positive signals for tide-surge interactions in $R_{surge}$ at Tanjong Pagar and Johor Baharu and in $R_{sum}$ at Geting when using the uniform distribution. Using the No-TSI distribution at locations that have similar frequency distributions such as Green Island in New Zealand (Costa et al., 2023), Newlyn in UK (Haigh et al., 2010), and several locations in China (Feng et al., 2019) may lead to a different conclusion regarding the existence of tide-surge interactions.

The existence of strong tide-surge interactions in this region discourages the use of the revised joint probability method for estimation of sea level extremes, which works well in locations with short records but tends to overestimate return levels at locations with strong tide-surge interactions (Arns et al., 2020; Olbert et al., 2013; Haigh et al., 2010; Tawn, 1992; Tawn and Vassie, 1989). The use of skew surge for the estimation of extremes may be more suitable than surge or residuals for this region (Williams et al., 2016).

**5 Conclusions**

We have introduced the No-TSI distribution to be used in determining the presence of tide-surge interaction. The No-TSI distribution can account for irregular tidal cycles that can lead to non-uniform sampling with respect to the number of hours from HW. Hence, the No-TSI distribution is generally not

a uniform distribution. When determining the presence of tide-surge interaction, the observed frequency
distribution should be compared to the No-TSI distribution instead of a uniform distribution.
Analysis of tide gauge records using the No-TSI distribution provides evidence of tide-surge
interaction at all seven tide-gauge locations along Singapore and the east coast of Peninsular Malaysia.
The observed interactions have a smooth spatial dependence along the coastline. During semi-diurnal
tidal cycles at the southernmost location of Tanjong Pagar, extreme residuals are mostly found around 5
hours before tidal HW with a much smaller number of extremes occurring after HW. Moving northwards,
similar patterns are found at Johor Baharu and Sedili, until we reach Tioman where extremes are mostly
found before HW but many extremes can also be found after HW. Northwards from Kuantan, most
extremes are found after HW. The strong tide-surge interaction in this region suggests that return levels
should be estimated using skew surge, not hourly residuals.
To investigate the contribution of tidal phase alteration, we proposed a semi-empirical model. We
used 10 m winds from ERA5 to estimate the effects of tidal phase alteration. At the six southern stations
of Tanjong Pagar, Johor Baharu, Sedili, Tioman, Kuantan and Cendering, we found that $R_{\text{phase}}$—the
residual component caused by the advancement of tidal HW due to surges—can significantly alter the
timing of extremes despite being responsible for less than 1% of the variance of the total residual. This
demonstrates the effects of tidal phase alteration on the timing of extreme residuals.
Our model has explored one of the underlying mechanisms behind tide-surge interaction but is
not designed to forecast water level or extreme events. A forecast model would require much more
accurate modelling of storm surge. The inclusion of other underlying mechanisms of tide-surge
interaction, such as the effect of tidal level on surge generation, would also be beneficial. Knowledge of
the interplay between tide-surge interaction and extreme sea levels can aid in the design of effective
strategies for coastal planning, risk assessment, and mitigation measures, and will benefit from more
comprehensive analyses. When studying extreme sea level in Southeast Asia, the relatively short length
of available tide gauge records poses a challenge, providing a focus for further research.

# Appendix A: Calculating $p$-value with bootstrapping

Here, our objective is to determine whether the *frequency distribution* obtained in Sect. 3.2 is drawn from the No-TSI distribution. To do so, we calculate the $p$-value: the probability of obtaining distributions from the No-TSI distribution that are at least as extreme as the *frequency distribution*. We can estimate this probability by considering the following: if we draw a random sample from the No-TSI distribution, the probability of drawing an outcome that is equally as probable or less probable than the *frequency distribution* is, by the definition of $p$-values, equal to $p$. Hence, we can obtain a good estimate of the $p$-value by drawing many samples from the No-TSI distribution and calculating the ratio of samples that are equally as probable or less probable than the *frequency distribution* to be drawn from the No-TSI distribution.

To obtain one bootstrap sample, following the notation in Sect 3.3, we draw one sample of size $n$ from the normalised No-TSI distribution $p_h$. To proceed, we need to calculate the probability of obtaining this specific outcome. Labelling this outcome as $\boldsymbol{k}$, $\boldsymbol{k}$ follows a multinomial distribution with probability given by $p_{\{\boldsymbol{k}\}} = \frac{n!}{\prod_h k_h!} \prod_h p_h^{k_h}$ where $k_h$ is the number of times (out of $n$ times) each possible $h$ is drawn. We can also compute $p_{\{\boldsymbol{k}^{(0)}\}}$, the probability of obtaining the *frequency distribution* from the No-TSI distribution, in the same way using the multinomial distribution. After obtaining 1,000,000 bootstrap samples and calculating their probabilities, we can achieve a good estimate of the $p$-value as

$$p = \frac{\text{\# of samples where } p_{\{\boldsymbol{k}\}} \leq p_{\{\boldsymbol{k}^{(0)}\}}}{1,000,000}.$$

*Code and data availability*. The hourly tide gauge data can be downloaded from the University of Hawaii Sea Level Center (UHSLC) at https://uhslc.soest.hawaii.edu/data/?rq (Caldwell et al., 2001). The UTide MATLAB functions can be downloaded from https://www.mathworks.com/matlabcentral/fileexchange/46523-utide-unified-tidal-analysis-and-prediction-functions (Codiga, 2011). The ERA5 hourly data can be downloaded from https://doi.org/10.24381/cds.adbb2d47 (Hersbach et al., 2018). The bathymetry data can be downloaded from GEBCO at https://download.gebco.net/ (GEBCO Compilation Group, 2023). The analysis code used to produce the figures and tables can be downloaded from https://doi.org/10.5281/zenodo.12721300.


*Author contributions*. ZYK: conceptualisation; data curation; formal analysis; investigation; methodology;
software; visualisation; writing – original draft preparation; writing – review and editing. BSG: conceptualisation;
investigation; methodology; writing – review and editing. DS: writing – review and editing. ADS:
conceptualisation; writing – review and editing. BPH: funding acquisition; supervision (supporting); writing –
review and editing. JD: funding acquisition; supervision (supporting); writing – review and editing. LYC: funding
acquisition; project administration; resources; supervision (lead); conceptualisation; investigation; methodology;
writing – review and editing.

*Competing interests*. The authors declare that they have no conflict of interest.

*Acknowledgements*. This Research is supported by the National Research Foundation, Singapore, and National
Environment Agency, Singapore under the National Sea-Level Programme Funding Initiative (Award No. USS-
IF-2020-3 and USS-IF-2020-1) and the Ministry of Education, Singapore, under its MOE AcRF Tier 3 Award
MOE2019-T3-1-004. Any opinions, findings, conclusions, or recommendations expressed in this material are those
of the author(s) and do not reflect the views of the National Research Foundation, Singapore, and the National
Environment Agency, Singapore. This work comprises EOS contribution number 606.

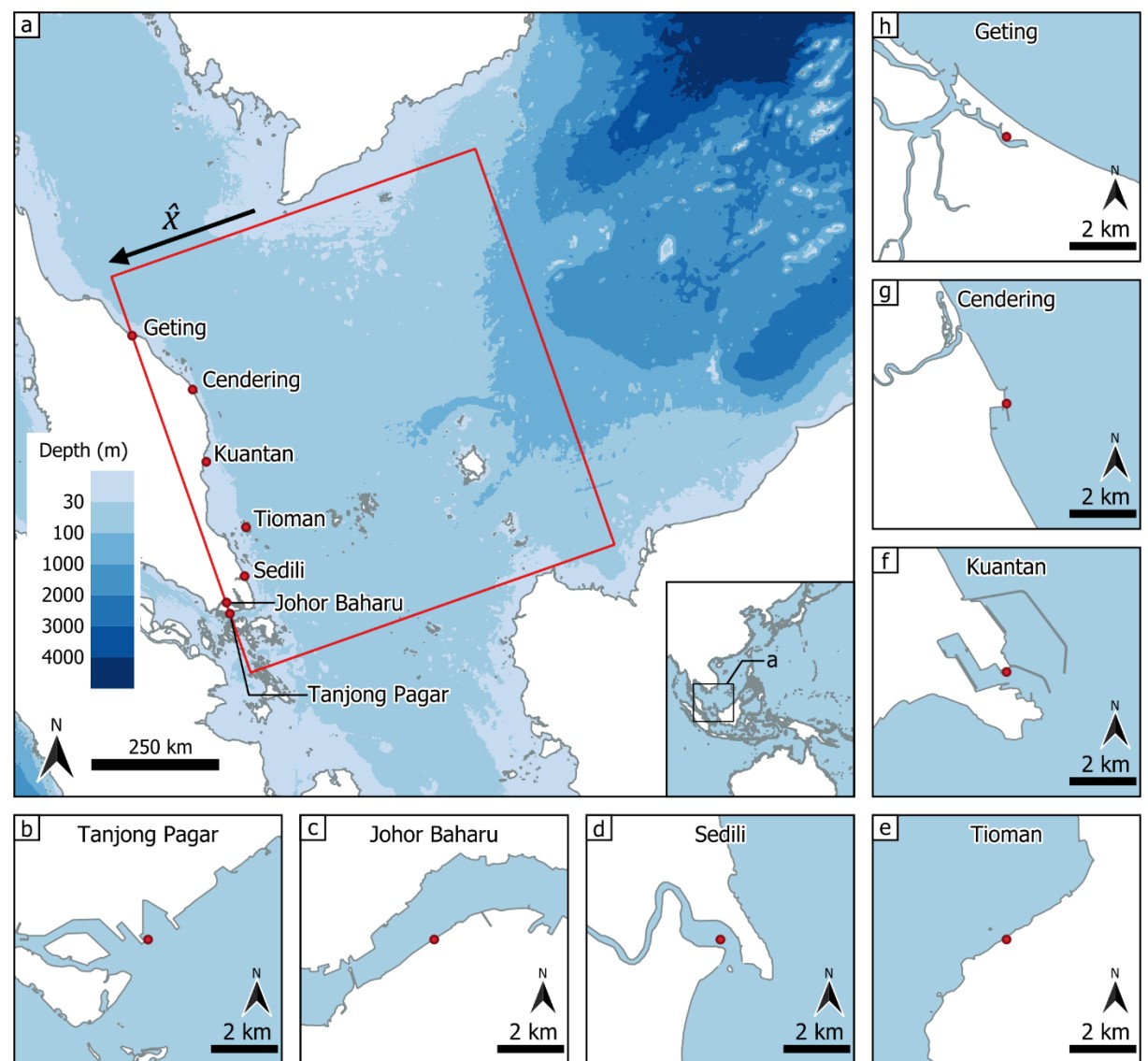

**Figure 1:** (a) Bathymetry of the region of interest of this study, obtained from GEBCO, the General Bathymetric Chart of the Oceans (GEBCO Compilation Group, 2023). The inset shows the location of this region within East and Southeast Asia. The seven tide gauge stations analysed are marked in red circles. The red rectangle denotes the region where 10 m winds and mean sea level pressure are considered when calculating $R_{\mathrm{surge}}$ (Sect. 3.4). The region is roughly a rectangle of 759 km by 833 km. The unit vector $\hat{x}$ used in Eq. (5)–(7) is shown in the figure. (b–h) Close up of the seven tide gauge locations (OpenStreetMap contributors, 2017).

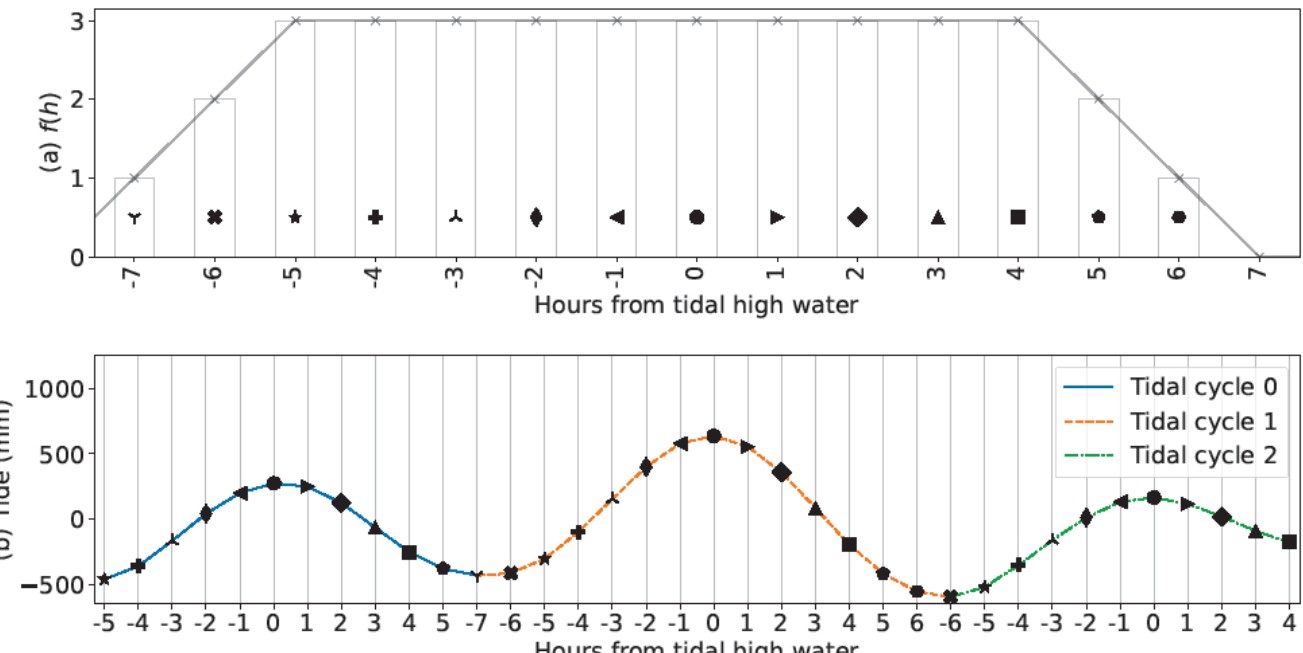

**Figure 2: An example of how the No-TSI distribution is obtained, using three tidal cycles at Tanjong**
**Pagar tide gauge station between 1 and 2 Jan 1989 (GMT). The number of hourly measurements**
**made at $h$ hours from HW are counted and denoted as $f(h)$, the No-TSI distribution before**
**normalization and scaling by the number of events (Sect. 3.3). In this example, if an extreme**
**residual can occur at any hour with equal probability, it will be 3 times more likely to happen at**
**HW than at 7 hours before HW. This is observed from (b) where $-5 \leq h \leq 4$ occur thrice, $h = -6$**
**and $h = 5$ occur twice, and $h = -7$ and $h = 6$ occur once, resulting from the three irregular tidal**
**cycles.**

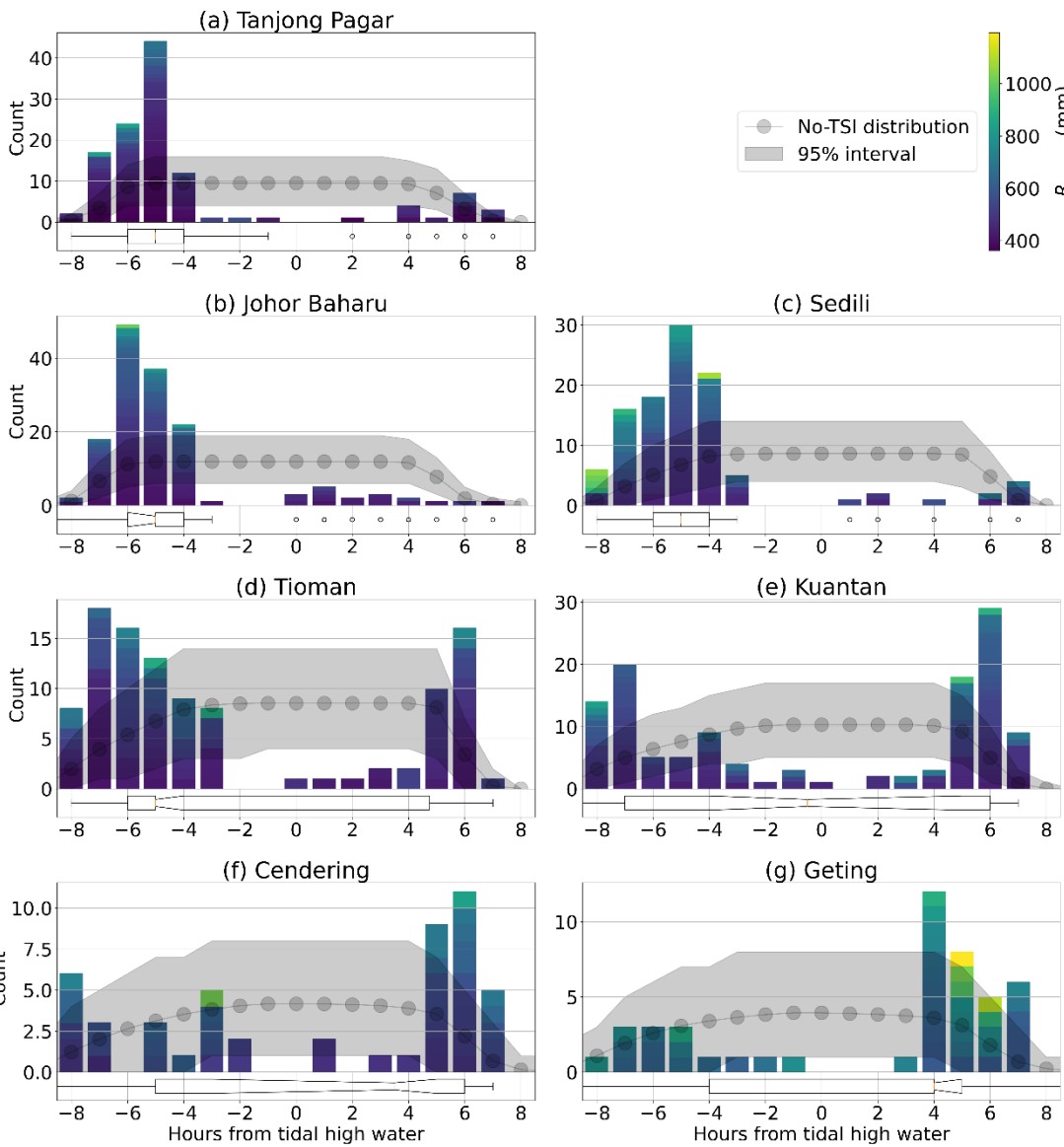

**Figure 3: Frequency distribution—the number of extreme events that have occurred at a certain number of hours relative to HW—of extreme residuals ($R_{\text{gauge}}$) during semidiurnal tidal cycles (Sect. 3.2). The plots are truncated at $\pm 8$ hours from HW. The frequency distribution is compared to the No-TSI distribution, shown in grey, to determine the presence of tide-surge interaction (Sect. 3.3). Summary statistics of the frequency distribution are shown using the horizontal notched box plot (Sect. 3.2). The whiskers of the box plot at (b) Johor Baharu, (e) Kuantan, (f) Cendering and (g) Geting extend beyond $\pm 8$ hours from HW, and their full extent is shown in Figure 4.**

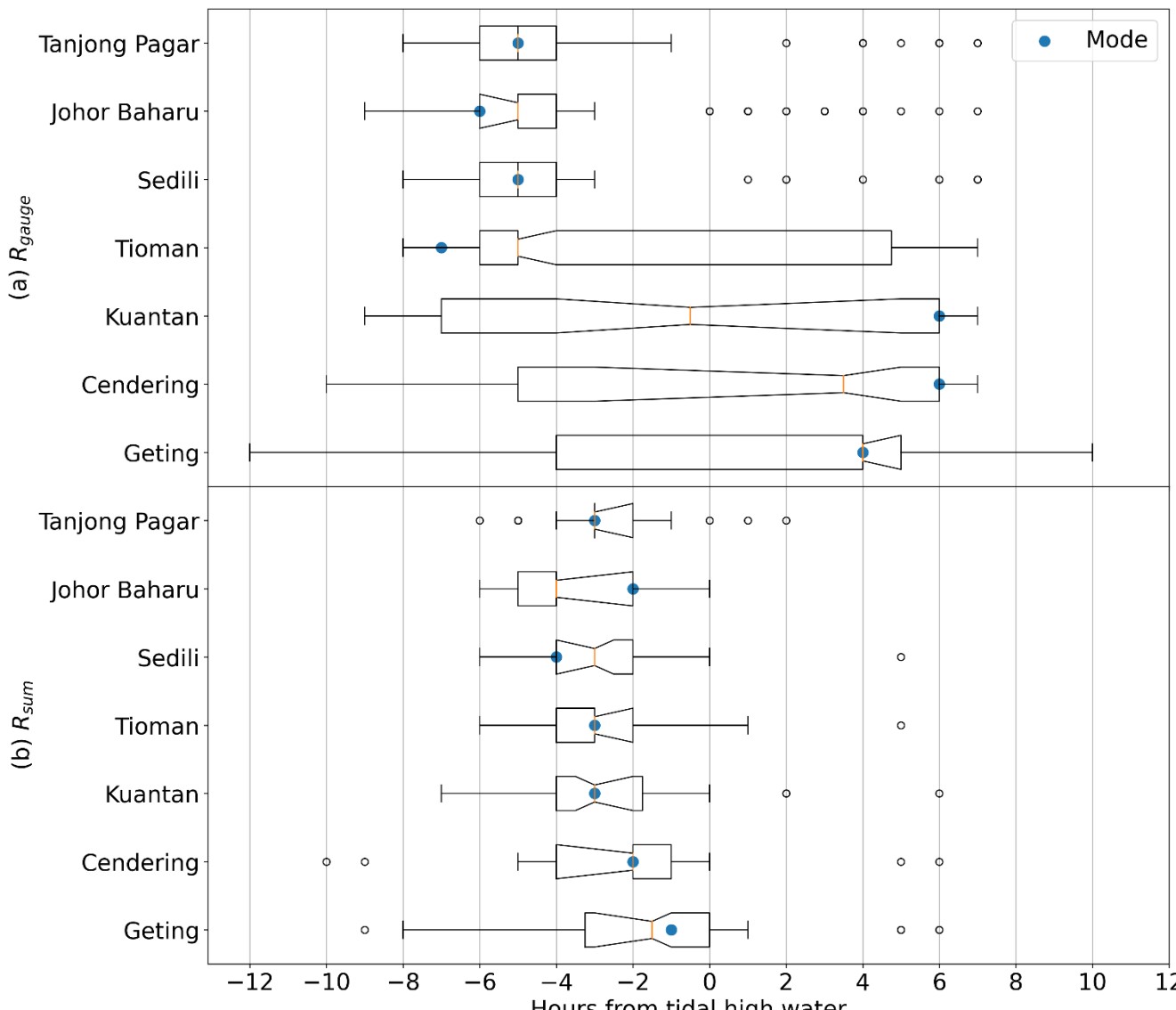

Figure 4: Compilation of the box plots in Fig. 3 and Fig. 5. Results presented in Fig. 3 on the timing of extreme residuals are compiled in subplot (a) and results presented in Fig. 5 on extreme values of $R_{\mathrm{sum}}$ are compiled in subplot (b). The box plots illustrate summary statistics of the distribution $k^{(0)}$ at each location, where orange lines indicate the medians, notches indicate the 95% confidence interval of the medians, blue circles indicate the modes, notched rectangles indicate the interquartile range (IQR), whiskers indicate a range that extends up to 1.5×IQR from the limits of the IQR, and black circles indicate outliers outside this range.

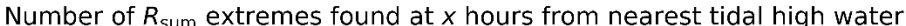

Figure 5: The frequency distribution for extreme values of $R_{sum}$ and the No-TSI distribution during semidiurnal tidal cycles, truncated at $\pm 8$ hours from tidal HW. Truncated horizontal notched box plots illustrate some summary statistics of the frequency distribution, and their full extents are shown in Fig. 4.

**Table 1: Data availability of the tide gauges used in this study. Completion rate is the percentage of usable hourly observations out of the duration of records. Usable rate is the percentage of usable observations subtracting 1-year moving averages (Sect. 3.1).**

| Location | Latitude | Longitude | Start | End | Years | Completion (%) | Usable (%) |
|---|---|---|---|---|---|---|---|
| Tanjong Pagar | 1.262 | 103.853 | 1988-01-01 | 2018-12-30 | 31.0 | 95.42 | 89.24 |
| Johor Baharu | 1.462 | 103.792 | 1983-12-19 | 2013-12-31 | 30.0 | 95.02 | 87.11 |
| Sedili | 1.932 | 104.115 | 1986-12-23 | 2015-12-09 | 29.0 | 98.08 | 98.08 |
| Tioman | 2.807 | 104.140 | 1985-11-13 | 2015-12-31 | 30.1 | 96.47 | 89.22 |
| Kuantan | 3.975 | 103.430 | 1983-12-22 | 2015-12-31 | 32.0 | 98.70 | 98.70 |
| Cendering | 5.265 | 103.187 | 1984-11-01 | 2015-11-30 | 31.1 | 96.58 | 90.58 |
| Geting | 6.227 | 102.107 | 1986-12-17 | 2015-12-31 | 29.0 | 99.14 | 99.14 |

**Table 2: Standard deviation and certain key quantiles of detrended water level, $X(t) - Z(t)$ , at the**
**seven tide gauges (mm).**

| Location | Standard Deviation | Min | Lower quartile | Median | Upper quartile | Max |
|---|---|---|---|---|---|---|
| Tanjong Pagar | 713 | -2011 | -539 | 45 | 578 | 1919 |
| Johor Baharu | 767 | -2267 | -575 | 67 | 596 | 2007 |
| Sedili | 569 | -1703 | -397 | 49 | 422 | 1843 |
| Tioman | 644 | -1984 | -436 | 6 | 457 | 2045 |
| Kuantan | 650 | -2050 | -440 | -27 | 446 | 2173 |
| Cendering | 525 | -1665 | -371 | -22 | 368 | 1843 |
| Geting | 321 | -937 | -238 | -26 | 217 | 1412 |


**Table 3: Summary of the tidal characteristics at the study locations including tidal range, four tidal**
**constituents ($K_1$, $O_1$, $M_2$, $S_2$), and the tidal form factor ($F$). Units, where applicable, are in metres**
**(m). The diurnal tidal range is the difference between mean higher high water and mean lower low**
**water, and is also referred to as the great diurnal tidal range or great diurnal range (NOAA, 2000).**
**Maximum tidal range is the greatest difference between higher high and lower low water within a**
**single day, and is much larger than the diurnal tidal range.**

| Location | Tidal Range | | $K_1$ | $O_1$ | $M_2$ | $S_2$ | $F$ |
| --- | --- | --- | --- | --- | --- | --- | --- |
| | Diurnal | Max | | | | | |
| Tanjong Pagar | 2.2 | 3.3 | 0.31 | 0.30 | 0.80 | 0.32 | 0.54 |
| Johor Baharu | 2.4 | 3.6 | 0.31 | 0.31 | 0.88 | 0.34 | 0.50 |
| Sedili | 1.8 | 2.8 | 0.35 | 0.31 | 0.56 | 0.16 | 0.91 |
| Tioman | 2.1 | 3.5 | 0.46 | 0.34 | 0.60 | 0.19 | 1.03 |
| Kuantan | 2.0 | 3.6 | 0.53 | 0.36 | 0.53 | 0.18 | 1.26 |
| Cendering | 1.5 | 2.7 | 0.49 | 0.31 | 0.30 | 0.12 | 1.90 |
| Geting | 0.8 | 1.4 | 0.25 | 0.13 | 0.17 | 0.08 | 1.56 |

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
