# Peer review of "Tide-Surge Interaction observed at Singapore and the east coast of Peninsular Malaysia using a Semi-empirical Model"

_EGUsphere, 2024_

## Referee Comment (RC1)

**REVIEWER COMMENTS**

Main comments:

The paper shows an innovative approach to evaluate whether tide-surge interactions occur when considering mixed-tide environments (when diurnal and semidiurnal tides happen at the same place). In addition, the paper brings an interesting approach to studying the wind-induced surge and the mechanism of mutual phase interaction between surges and tides. Overall, the paper brings high-quality research, and it is well-written. However, it must be improved before publication. My main comments are:

**Introduction:** I suggest the authors clarify/highlight why tide-surge interactions are important in the study area. Have there been any remarkable extreme events in the past? Is the area prone to typhoons? What about the geographical setting? The authors have mentioned some aspects of it, but looking at the mapping area (Figure 1), I can easily spot more characteristics that can be interesting to mention, like the presence of islands and a land constriction further south. What is the geographical setting of each tide gauge location?

**Data and Methods.** Some passages in the section are confusing. Here are some general comments:

1. Before the subsections start, I suggest explaining the structure of the method (TSI identification, semi-empirical model), like a "road map." For instance, generally state what the main databases used are and the main steps of the methodology. It should be a short paragraph.
2. Some terms should be better defined, such as "clusters" and "uniform distribution." Also, why use 168 hours to decluster storms? The authors gave an explanation that is not sufficient, in my opinion. They should provide a physical explanation for that. For instance, what is the main duration of storms in the area?
3. The bootstrapping method is a bit hard to follow, as I am not familiar with it. The authors should improve the explanation by introducing more details, further explaining some basic concepts in the text, or making it more visual with some figure or scheme. That would help readers not used to the methodology understand.
4. Why did the authors not consider the storm surge due to the horizontal gradient of the atmospheric sea level pressure?

**Results and discussion.** Here is where the paper could improve the most. The authors made some interesting analyses but explored little of that in the discussion. The authors should further discuss the link between the statistical analysis, semi-empirical model and physical processes. Little was discussed in terms of physical interpretation. Bellow I suggest some discussion points:

1. What are the limitations of the semi-empirical model? Does the semi-empirical approach fully consider the local geographical setting and physical processes? Explain why not in more detail and explain how this affects the results. Try to discuss the results while considering each tide gauge's local characteristics.
2. How could the research's findings be useful worldwide? What are the mixed tide regions around the globe? Comparing the manuscript results with some global studies like Arns et al. 2020 and focusing on analysing mixed tide regions could be a good connection point in your discussion.
3. How are data-driven and numerical models' approaches limited to model TSI, and how can the proposed semi-empirical model provide a valuable option for this type of analysis?
4. (optional) Have the authors considered mentioning/discussing the skew-surge approach? How would the mixed tides regime affect methods using skew-surge?
5. The authors mentioned that considering TSI is important; however, they did not explain the main limitations/concerns of not considering it.

**Figures**. Please pay attention to referencing the panels that each figure contains in the text. I do not know exactly the limit on the number of figures in the main text, but I suggest the authors consider moving some figures from the Appendix to the main text.

Specific comments:

1. **Line 33.** Please define storm surge; I do not think this statement is true. Actually, storm surges (only atmospheric forcing) are larger when the depth is lower (ebb to low tide). That happens because the effects of shallow water are stronger. For instance, the contribution of wind set-up is larger in shallow waters or low tide. Thus, saying that storm surge is larger when high tide is not true. Some authors define storm surge as the sum of non-tidal residuals and astronomical tides. However, the combined water level resulting from non-tidal residuals and astronomical tides is often called storm tide (e.g., Stephens et al., 2020).

2. **Line 84.** What is the vertical reference? Mean sea level? Does the 2.7-3.6 m range refer to the high tide or tide amplitude?

3. **Line 85.** Please change the sentence "Hydrodynamical processes have a strong influence on the water levels at the seven tide gauges" to "Hydrodynamical processes have shown a strong influence on the water levels at the seven tide gauges analysed in this study".

4. **Line 86.** Which hydrodynamical processes? Please be more specific. Also, detailed the local geographical setting. Are the tide gauges on the open coast, inside bays, sheltered by an island?

5. **Line 97:** "Duration and skewness of the tidal cycle." Skewness is not defined in the text. Is it tidal asymmetry? Moreover, are the different durations and skewness a local characteristic? It should be better explained.

6. **Lines 98 -105.** It seems an interesting approach. However, there is too much information in a single paragraph, where the authors add new terms but do not define them properly. I suggest simplifying or omitting some of the content and keeping it to the methods section. I suggest the focus here to be why we cannot use existing methods to identify TSI in mixed tidal regions. The authors said that tide cycles do not have the same duration. Why and how would that interfere with the TSI assessment? The answer is already there, partially. However, the authors must make it clearer.

7. **Line 141.** Where did this information come from? Would the authors have any references to cite?

8. **Lines 142-144.** The paragraph does not follow a logical order. Why are the authors talking about tide periodicity in this paragraph? I do not see a clear connection between the last sentence and the two first sentences of the paragraph. Please make it clearer or separate it into a new paragraph.

9. **Line 157.** Please define clusters. Expressions like "Longlist" and "shortlisted" should not be used.

10. **Line 161.** I found this time period quite large. The authors' use of this needs to be explained better. Which local characteristics have guided the authors to set such a period? Other works in the field usually use a 3-day period.

11. **Line 187.** A uniform distribution is not the most suitable option in which context? Please specify. Also, uniform distribution of what exactly? "Uniform distribution" should be properly defined.

12. **Line 201-214**. I found this paragraph hard to follow, even with the material in the appendix. Please see my notes on "General comments" (Data and Methods, 3).

13. **Line 210.** I suggest adding a short explanation about what a probability mass function is.

14. **Line 264-266.** The sentence "Plotting… at all seven tide gauges reveal a correlation of 0.7-0.8" shows a result. Please remove it and keep this information in the Results section.

15. **Line 267.** Why do the authors need to fit a linear regression to find k? k is given by constants pair, Cd,p,g,D.

16. **Line 274.** Should Ltide be a "tidal wave not under or free of the influence of Rwind"?

17. **Line 285.** Figure 3 should be described better. For instance, instead of putting Figure 3 in parenthesis, the sentence should start describing the figure. Figure 3 has 7 panels (a-g); why did the authors not mention them in the main text? This happens to most figures with several panels on them. There are no references to the different panels in the text. Please review all your figures and change them accordingly.

18. **Line 290-298**. From a physical point of view (e.g. local geographical set, physical processes), what could explain these differences in time of extreme residual occurrence? That is a good discussion point.

19. **Line 299.** The bootstrapping method should be explained better. Why is it used? Detailed information about it should be described in methods, not results.

20. **Line 311.** Which modes? What figure are you referring to? Are you talking about tidal components, like M2 and S2? Or are you talking about the bins at your histogram of frequency of occurrence? Where did you define modes previously in the text? Is that the statistical mode, e.g., the most frequent value of a distribution?

21. **Line 316.** Please reference the different panels in Figure 4.

22. **Line 331-338.** Please provide some possible physical explanations for the results.

23. **Line 340-343.** This information repeats what was already said in the methods section. Please remove it from methods and keep it here.

24. **Line 344-345.** Are you comparing the timing of Rwind and No-TSI's timing (phase difference)? Or the distribution at which the extremes occur? It is not clear what you are comparing exactly. Please rewrite the sentence to make it clear.

25. **Line 351.** Figures S6-S9 need a better legend. Please replace "counts" with "Rwind" when applicable.

26. **Line 355.** Figures S10-S13 need a better legend. Please replace "counts" with "Rphase" when applicable.

27. **Line 366-373.** Join this paragraph to the last one.

28. **Line 374.** The authors are adding new methods to the results. This procedure should be added to methods, not here. Also, the purpose of Rsum is to compare with Rgauge. Please make it clearer.

29. **Line 378. "This can also be seen in Figure 4…".** Actually, in Cendering, the median and mean are quite similar. Please correct it.
30. **Line 380**. In the sentence " We find that…", where is the result shown? In which figure?

31. **Line 381-384.** This is a good discussion point. Why do diurnal tides not experience the TSI? Please develop a physical explanation for that.

32. **Line 389-394.** I do not understand the connection between the last sentence and the rest of the paragraph. It is not clear what the point of discussion is. This last sentence should be the start of your paragraph (topic sentence), and your results (first two sentences of the paragraph) should support your idea. Please rewrite the paragraph to make your ideas and discussion clearer.

33. **Line 395.** Please Remove "however".

34. **Line 395.** Should I compare Rgauge and Rsum? If so, this needs to be better explained in the methods section and in the text. Which figures should I look at? Please refer to the text.

35. **Line 402 – 406.** Interesting discussion point; however, the authors should develop it further. What could you change in your model to account for these limitations? Here, you should talk about why your model is limited, how you could improve it, and reference works that may have done it somehow. The authors cited 2 references but did not give much detail and explore it further. I could see you used a constant depth to obtain Rwind. You should maybe account for shallower depths and several values. For instance, could you repeat the process by considering different depths? For example, you could calculate Lwind and Rwind from bathymetry 60m-40, then after 40-20, 20-10m... Do you think that would make a difference? Also, should we take different wind directions into account? Another point would be: Are there any local wind observations? You could use it instead of ERA5. Maybe some conceptual figures could be placed here to help explain mutual phase alteration, etc.... Another point is to discuss the other main mechanisms of tide surge interaction: advection, shallow water effects, and bottom friction. Are all your tide gauges located on the open coast? Or is any of them at a bay or harbour/estuary? That could have a massive impact on your results. Please see general comments on other important discussion points.

36. **Line 408.** I believe a similar approach was used by Arns et al. 2020.

37. **Line 411: "No-TSI distribution should be used instead of a uniform distribution." I propose you perform a classification using** a uniform distribution of your data and see how it performs compared to your approach. This would illustrate clearly what you want to show. It could be for only 1 or 2 sites. This would give you a good discussion point in the discussion section.

38. **Line 423: "** We found the residual component caused by advancement of tidal HW**."** Does the wind cause this? Please rewrite the sentence.

39. **Lines 430-433. Please move this sentence to the discussion section. There are usually no citations in the conclusion section**.

---

## Referee Comment (RC2)

**Tide-Surge Interaction near Singapore and Malaysia using a Semi-empirical Model**
Zhi Yang Koh, Benjamin S. Grandey, Dhrubajyoti Samanta, Adam D. Switzer Benjamin P. Horton, Justin Dauwels, Lock Yue Chew

The authors investigate the tide-surge interaction near Singapore and Malaysia based on observation data and with a semi-empirical model.

They found tide-surge interaction at all studied locations based on observations. They conclude that the tidal phase could change but with minimum impact on high water. Their semi-empirical model could not simulate the tide-surge interaction.

The paper is well written but there are a lot of unclear points for me.

For whom, do they have written this paper?

Why are the tide-surge interactions so important for this region if there is only a minimum change in water level. What does it mean e.g. for coastal protection or warning systems?

There is no time-series of water level including the diurnal semi-diurnal tides from the different locations, e.g. for one or two weeks. The locations are so different in their water levels.

How good are the observation data? Is there a trend in the observation data? Are there changes in tidal range at the locations over the investigated period?

Are there changes at the gauges, e.g. land subsidence, sea-level rise?

I'm wondering that data from a global database is used and no local data directly from the gauges (authorities) with e.g. 15-20 min resolution. Hourly observations could miss the low and high peaks.

Do the authors test another tide model? How good is the tide model compared to the observation during calm wind situations? How do the authors avoid making mistakes in the phase shift of the tides?

All results of the semi-empirical model are in the supplements. Why?

In general text and figures could be put more into context.

How high are the observed extreme water levels? And are they dangerous for the region?

Would you mind putting the bathymetry around the locations in table 1 or a higher resolution at the coast in figure 1. Is it steep or flat at the coastal locations?

The influence of the bathymetry on surges is a little bit too short in the text.

More specific questions and comments:

Are there regional hydrodynamic models available for this region to simulate surges and tide-surge interaction, too? In line 72 to 79, you mentioned already some paper.

Line 82: What kind of wave?

At the end of the Introduction a roadmap of the paper would be nice.

Line 131 to 135: Did you test other tide models?

Line 157- 162: How long is a storm in this region? Can two extreme storms occur within seven days?
Line 193 what are the differences between the specific locations? Unfortunately, I am not familiar with this region.
Line 201 -214 ???
Line 233-234 also for extreme water level? Is the wind direction always the same?
Line 237, what does it mean masking out the land? Are the locations now at different regions?
Line 243: How do you estimate the influence of bottom friction on the wind surge process with a depth of 58 m?
Line 262: The mean of wind speed over a large area of 759km x 833 km is probably different to wind speed with a spatial resolution of ~31km of ERA5. Please comment!

Why do the authors not use for every location their own wind speed and -direction in their semi-empirical model? ERA5 has a resolution of ~31km. I'm wondering that there is no hint about wind direction.

A storm tide is the sum of tides, wind surge, external surge, the inverse barometer effect, and non-linear tide–surge interactions. I miss the inverse barometer effect in your paper. (Please revise your definition in line 33-34).

Results and Discussion

I am wondering that I have not seen any box plot of the water level and the residuals for every location.

How big is the difference in water level between $99^{th}$ and $95^{th}$ (or $90^{th}$) percentile. Is the $95^{th}$ percentile closer to the high-water events than the $99^{th}$ percentile or is it the same. Are $90^{th}$ percentile events more dangerous because, they can occur during high tide.

For Figure S2, I would prefer the same axis length and it should in the main part.

S2 to S19 are in the supplements.

Conclusions

The authors provide a lot of statistics, but what do the results mean for decision-makers.

In summary, the paper needs some more investigations and clearer physical explanations.

---

## Author Comment (AC1)

Response to referee comments on

**"Tide-Surge Interaction near Singapore and Malaysia using a Semi-empirical Model".**

Zhi Yang Koh et al.

11 June 2024

**Introduction**

We thank the two referees for their detailed and constructive feedback on our manuscript. We respond to each comment (quoted in *green italics*) below. These comments have led us to reconsider our assumptions, improve our methodology, and expand our analysis, culminating in a significantly improved manuscript.

We have improved upon our methodology by including the height of residuals in our histograms, and by accounting for spatial variability of wind and bathymetry and the inverse barometer effect in our semi-empirical model. This has improved our model residuals and allowed us to identify spatial patterns in the residual heights. In our discussion, we now relate our results to physical explanations such as the geographical setting and the tidal form factor.

The manuscript has been substantially revised due to the above changes.

**Referee #1**

**Main comments**

*The paper shows an innovative approach to evaluate whether tide-surge interactions occur when considering mixed-tide environments (when diurnal and semidiurnal tides happen at the same place). In addition, the paper brings an interesting approach to studying the wind-induced surge and the mechanism of mutual phase interaction between surges and tides. Overall, the paper brings high-quality research, and it is well-written. However, it must be improved before publication. My main comments are:*

Thank you for your overall assessment and for your comments, which have helped us to improve the paper.

**Introduction**

*1.       I suggest the authors clarify/highlight why tide-surge interactions are important in the study area.*

Strong tide-surge interactions are expected at shallow regions such as the Sunda Shelf. The expanse of this shallow region facing the Peninsular Malaysian east coast also allows tidal phase alteration, one mechanism of tide-surge interaction, to develop before reaching the coast. Understanding this mechanism would improve water level prediction for operational forecast of sea level and aid development of statistical estimation techniques of sea-level extremes for

coastal protection design purposes. We have added these points to the introduction (lines 44–46, 49–50, 89–91) and discussion (lines 546–550) of the revised manuscript.

*2.      Have there been any remarkable extreme events in the past? Is the area prone to typhoons?*

Storm surges occur along the coast throughout the year, reaching 0.35–1.0 m at Geting, the northernmost location of our study region (Abd Razak et al., 2024). The northern part of the Peninsular Malaysian east coast is more prone to stronger storm surges due to tropical depressions (Mohd Anuar et al., 2023). In the southern part of the coast, a surge of about 1.0 m caused by Typhoon Vamei in 2001 was recorded (Mohd Anuar et al., 2018). Several storm surge events in Singapore have also been documented since 1974 (Luu et al., 2016), particularly during the winter monsoon period. We have added this information to the revised manuscript in lines 100–105.

*3.      What about the geographical setting? The authors have mentioned some aspects of it, but looking at the mapping area (Figure 1), I can easily spot more characteristics that can be interesting to mention, like the presence of islands and a land constriction further south.*

The geographical setting is as follows. The seven gauges are located within the Sunda Shelf, which has a typical depth of 40–80 m. The eastern coastline of Peninsular Malaysia faces the South China Sea and is exposed to strong seasonal monsoon winds resulting in larger surges (Mohd Anuar et al., 2020). To the north of our study region lies the Gulf of Thailand. To the south of Singapore, the southernmost location of our study region, lies part of the Riau Islands, Sumatra, the Karimata Strait, and the Java Sea. The Malacca Strait is west of Singapore and leads to the Indian Ocean. The diurnal and semidiurnal tides that propagate from the South China Sea and the Indian Ocean respectively drive the complex mixed tides in Singapore and the Southern parts of the study region (van Maren and Gerritsen, 2012). At Singapore, the diurnal tides that propagate from the South China Sea get further amplified from reflection against the east coast of Sumatra (van Maren and Gerritsen, 2012). We have added these brief descriptions of the region to lines 89–99 of the revised manuscript.

*4.      What is the geographical setting of each tide gauge location?*

Of our seven tide gauges, Tanjong Pagar and Johor Baharu are located within straits, Sedili at a river mouth, Tioman between Tioman island and mainland Peninsular Malaysia sheltered from the South China Sea, Kuantan at the mouth of a man-made bay, Cendering on the open coast and Geting sheltered inside a bay (Caldwell et al., 2001; OpenStreetMap contributors, 2017). We have added this brief description of the tide gauge locations to lines 153–156 of the revised manuscript. This is also mentioned in specific comment 4.

**Data and Methods**

*Some passages in the section are confusing. Here are some general comments:*

*1.      Before the subsections start, I suggest explaining the structure of the method (TSI identification, semi-empirical model), like a "road map." For instance, generally state what*

*the main databases used are and the main steps of the methodology. It should be a short paragraph.*

We have added a short paragraph to the beginning of Sect. 2 that outlines the general methodology (lines 139-147).

*2.     Some terms should be better defined, such as "clusters" and "uniform distribution."*

We have now clarified the meaning of several technical terms in response to the following specific comments:
Specific comment 1: Storm surge
Specific comment 2: Tidal amplitude
Specific comment 5: Tide duration and skewness
Specific comment 9: Longlist and shortlist (removed usage of these two terms)
Specific comment 11: Uniform distribution
Specific comment 13: Probability mass function
Specific comment 20: Modes

*Also, why use 168 hours to decluster storms? The authors gave an explanation that is not sufficient, in my opinion. They should provide a physical explanation for that. For instance, what is the main duration of storms in the area?*

We choose a threshold of 1 week (168 hours) to reduce the odds of double counting long-lasting surges, as monsoon surge events can last between 1 and 5 days in this region (Meteorological Service Singapore; Tkalich et al., 2009). We also found statistically significant autocorrelation in the $R_{gauge}$ time-series at time lags shorter than 168 hours. A declustering threshold of 168 hours still retain sufficient observations for our analysis. We have added this explanation to the revised manuscript in lines 216-221. This is also mentioned in specific comment 10.

*3.     The bootstrapping method is a bit hard to follow, as I am not familiar with it. The authors should improve the explanation by introducing more details, further explaining some basic concepts in the text, or making it more visual with some figure or scheme. That would help readers not used to the methodology understand.*

We have improved on the explanation of how the No-TSI distribution is obtained and how the bootstrap procedure is implemented through the following:

- No-TSI (Sect. 2.3): we have made further reference to Fig. 2, made some edits and removed unnecessary details to make the procedure easier to understand.
- Bootstrapping (Appendix A): we have rewritten the appendix, removing unnecessary details to make the procedure easier to understand.

See also specific comments 12 and 19.

*4.     Why did the authors not consider the storm surge due to the horizontal gradient of the atmospheric sea level pressure?*

Our semi-empirical model aims to provide a first-order estimate of the effects of tidal phase alteration. Hence, we opted to reduce the complexity of our model and did not incorporate atmospheric sea level pressure.

However, in response to the reviewer comments, we have modified the model to account for sea level pressure. This has slightly improved our model surge, $R_{\text{sum}}$, by $R^2$ (coefficient of determination) of 0.0017–0.0083 when compared to $R_{\text{gauge}}$, except at Johor Baharu where $R^2$ decreased by 0.0015.

**Results and discussion**

*Here is where the paper could improve the most. The authors made some interesting analyses but explored little of that in the discussion. The authors should further discuss the link between the statistical analysis, semi-empirical model and physical processes. Little was discussed in terms of physical interpretation. Bellow I suggest some discussion points:*

*1. What are the limitations of the semi-empirical model? Does the semi-empirical approach fully consider the local geographical setting and physical processes? Explain why not in more detail and explain how this affects the results. Try to discuss the results while considering each tide gauge's local characteristics.*

We mention the following limitations in the revised manuscript (lines 512–534) to address these insightful comments:

- Shallower water during low tides can result in higher surges and cause mutual phase alteration.
- Bottom friction and momentum advection were not explicitly accounted for.
- We could not account for complex tidal currents between the Pacific and Indian Oceans.
- Tide gauge-specific details that we could not account for include tide-surge-river interactions at Sedili, hydrodynamics due to Tioman Island, and the impact of the Gulf of Thailand on the hydrodynamics at the northern gauges.
- Use of dynamical models or a partial differential equation solver could potentially address the above shortcomings to some extent. However, such approaches are beyond the scope of the current work.

*2. How could the research's findings be useful worldwide? What are the mixed tide regions around the globe? Comparing the manuscript results with some global studies like Arns et al. 2020 and focusing on analysing mixed tide regions could be a good connection point in your discussion.*

We have added further discussion on how our results can be useful in other regions of the world, and how it can inform future research and real-world applications (lines 535–550). The discussion points are, in summary:

- Use of the No-TSI distribution can improve hypothesis testing globally.
- The semi-empirical model could potentially be used at locations with similar local characteristics, including a shallow shelf over which wind-driven surges develop and tides propagate.

- Our results indicate that using skew surge may be more suitable than surge when estimating extremes in this region.

*3.    How are data-driven and numerical models' approaches limited to model TSI, and how can the proposed semi-empirical model provide a valuable option for this type of analysis?*

Numerical hydrodynamic models resolve the physical processes in more detail and hence are more complex and computationally intensive. While it does not replace hydrodynamical models, our semi-empirical model offers theoretical insight by providing a clear and transparent explanation of tidal phase alteration and its impact at a low computational cost. We have added this point to lines 531–534 of the revised manuscript.

*4.    (optional) Have the authors considered mentioning/discussing the skew-surge approach? How would the mixed tides regime affect methods using skew-surge?*

Thank you for your suggestion. We have added some discussion on skew-surge to the revised manuscript stating that the use of skew-surge would be preferred over non-tidal residuals when estimating extreme water level in this region due to the strong tide-surge interaction (lines 546–550). As skew surge is a measure of height and not timing, the mixed tide regime in this region may not require any special treatment.

*5.    The authors mentioned that considering TSI is important; however, they did not explain the main limitations/concerns of not considering it.*

Kurniawan et al. (2015) and Tay et al. (2016) found significant contribution in the order of 10 cm to non-tidal residuals by tide-surge interactions. Hence, consideration of tide-surge interactions is important in operational forecasting and statistical estimation of return levels. We have added this point to lines 529–531 of the revised manuscript.

**Figures**

*1.    Please pay attention to referencing the panels that each figure contains in the text.*

We now carefully reference the panels/sub-figures of Fig. 3, 4 and 5 directly whenever relevant.

*2.    I do not know exactly the limit on the number of figures in the main text, but I suggest the authors consider moving some figures from the Appendix to the main text.*

To clarify, even though the figures are currently displayed after the Appendix, Fig. 1–5 and Tables 1–2 will be displayed within the main text of the final manuscript.

To keep the manuscript reader-friendly and concise, we choose to keep less important figures in the supplementary material. This is especially considering that most figures in the supplementary material takes up almost one whole page. The five figures in the main text are the most important for illustrating our methodology and presenting our key results.

**Specific comments**

*1.    Line 33. Please define storm surge; I do not think this statement is true. Actually, storm surges (only atmospheric forcing) are larger when the depth is lower (ebb to low tide). That happens because the effects of shallow water are stronger. For instance, the contribution of*

*wind set-up is larger in shallow waters or low tide. Thus, saying that storm surge is larger when high tide is not true. Some authors define storm surge as the sum of non-tidal residuals and astronomical tides. However, the combined water level resulting from non-tidal residuals and astronomical tides is often called storm tide (e.g., Stephens et al., 2020).*

Thank you for this helpful comment. We meant that high tides would amplify the effects of storm surges (not the surges themselves) if they happen to coincide. We have addressed this ambiguity and have defined storm surge, rewriting the line to: "Storm surges are rises in sea level brought about by low atmospheric pressure and strong winds acting on the sea surface (Pugh and Woodworth, 2014a). The destructive effect of large storm surges can be amplified by high tides if the surge coincides with high tide to produce large storm tides (Stephens et al., 2020, Gregory et al., 2019), increasing the risk of coastal flooding and threatening lives and properties (Pugh and Woodworth, 2014c)."

*2.        Line 84. What is the vertical reference? Mean sea level? Does the 2.7-3.6 m range refer to the high tide or tide amplitude?*

We have now clarified in the text that the 2.7-3.6 m range refers to tidal range. Hence, a vertical reference (e.g. mean sea level) is not directly relevant here.

*3.        Line 85. Please change the sentence "Hydrodynamical processes have a strong influence on the water levels at the seven tide gauges" to "Hydrodynamical processes have shown a strong influence on the water levels at the seven tide gauges analysed in this study".*

Thank you for your comment; we have made the suggested edit.

*4.        Line 86. Which hydrodynamical processes? Please be more specific.*

We have added that the main contributors are "regional wind forcing during the seasonal monsoons".

*Also, detailed the local geographical setting. Are the tide gauges on the open coast, inside bays, sheltered by an island?*

We briefly describe the tide gauge locations in lines 153–156 of the revised manuscript (also referenced above in response to main comment 4 on the Introduction section).

*5.        Line 97. "Duration and skewness of the tidal cycle." Skewness is not defined in the text. Is it tidal asymmetry? Moreover, are the different durations and skewness a local characteristic? It should be better explained.*

Yes, Skewness refers to tidal asymmetry and these are local characteristics. We have revised the text to clarity this (lines 120-121).

*6.        Lines 98-105. It seems an interesting approach. However, there is too much information in a single paragraph, where the authors add new terms but do not define them properly. I suggest simplifying or omitting some of the content and keeping it to the methods section. I suggest the focus here to be why we cannot use existing methods to identify TSI in mixed tidal regions. The authors said that tide cycles do not have the same duration. Why and how would that interfere with the TSI assessment? The answer is already there, partially. However, the authors must make it clearer.*

Thank you for this suggestion. We have rewritten these lines to focus on the above points as per your suggestions: "The tidal asymmetry causes the rising and falling tide to not always be 6.5 hours long, and the variable tidal cycle durations means that 13 hourly bands are not always sufficient to fully characterise tide-surge interactions especially at locations with diurnal or mixed tidal regimes where one tidal cycle can be up to 25 hours long. Hence, our key modification to the existing methodology is to replace the uniform distribution with a "No-TSI distribution" as the null distribution. We also use an exact statistical test for hypothesis testing instead of the chi-square test. In addition, we propose a simple approach to compare the tide-surge interaction across locations that have different tidal characteristics (diurnal, mixed, semidiurnal)."

*7.      Line 141. Where did this information come from? Would the authors have any references to cite?*

Tidal range was from our results of UTide (Unified Tidal Analysis and Prediction Functions). We also added some references that have similar findings (lines 188–189). A tidal range of 3.6 m (amplitude of 1.8 m) causes water level to deviate roughly ±1.8 m from the mean level. If the depth is 40 m, then this corresponds to a deviation of 4.5 %.

*8.      Lines 142-144. The paragraph does not follow a logical order. Why are the authors talking about tide periodicity in this paragraph? I do not see a clear connection between the last sentence and the two first sentences of the paragraph. Please make it clearer or separate it into a new paragraph.*

We have rewritten these lines to improve the logical flow (lines 189–199): "The tides at a location can be categorised into diurnal, semi-diurnal, or mixed using the tidal form factor…". We decide to refrain from using 'periodicity' for better clarity. This sentence was meant to highlight the tidal form factor F as a common method used to classify tides at a location as diurnal or semidiurnal (hence 'periodicity').

*9.      Line 157. Please define clusters. Expressions like "Longlist" and "shortlisted" should not be used.*

We have rewritten these lines to avoid using 'longlist' and 'shortlist', and properly defined a cluster (lines 210–214).

*10.      Line 161. I found this time period quite large. The authors' use of this needs to be explained better. Which local characteristics have guided the authors to set such a period? Other works in the field usually use a 3-day period.*

We provide additional justifications for our choice in lines 216–221 of the revised manuscript (also referenced above in response to main comment 2 on the Data and Methods section).

*11.      Line 187. A uniform distribution is not the most suitable option in which context? Please specify. Also, uniform distribution of what exactly? "Uniform distribution" should be properly defined.*

The uniform distribution is not the most suitable option when tides are not always semi-diurnal and symmetric. To properly define 'uniform distribution' and specify when it is not the most suitable option,  we have rewritten this line: "In existing studies, the null distribution is a

uniform distribution with a value of $n/13$ at $h = -6, -5, ..., 0, ..., 5, 6$ where $n$ is the total number of extreme events (Horsburgh and Wilson, 2007; Haigh et al., 2010; Antony and Unnikrishnan, 2013; Feng et al., 2019; Costa et al., 2023). It assumes that tidal cycles are always 13 hours long (i.e. semi-diurnal) and HW is always at the mid-point of this cycle (i.e. tides are always symmetric). Hence, if there are $n$ extreme events, we expect $n/13$ events to happen at each hourly band if extreme events are equally likely to occur at any stage of a tidal cycle. However, as tides are not always semi-diurnal and symmetric, the uniform distribution is not the most suitable null distribution to represent the null hypothesis."

*12.     Lines 201-214. I found this paragraph hard to follow, even with the material in the appendix. Please see my notes on "General comments" (Data and Methods, 3).*

We have revised the text for better clarity of the revised manuscript (Sect. 2.3, Appendix A; also see our response to main comment 3 on the Data and Methods section).

*13.     Line 210. I suggest adding a short explanation about what a probability mass function is.*

We have added a short description: "Note that $p_h$ is a probability mass function over the domain $h$: a function that tells us the probability of an extreme event happening at $h$ is $p_h$."

*14.     Line 264-266. The sentence "Plotting… at all seven tide gauges reveal a correlation of 0.7-0.8" shows a result. Please remove it and keep this information in the Results section.*

We have removed this sentence. A similar sentence in the Results section is unchanged.

*15.     Line 267. Why do the authors need to fit a linear regression to find k? k is given by constants pair, Cd, p, g, D.*

The values of $\rho, g, D$ are precisely known and easily obtainable. However, $C_d$ is problematic and depends on basin characteristics. Precise calculation of $C_d$ is not the primary interest of our study so we opt to simply fit it. It is often used as a final tuning parameter in non-tidal barotropic models (Zweers et al., 2012, Kurniawan et al., 2015). We have added this explanation to the methodology section 2.4. Note that the numerical values of $g$ and $D$ are used in a later part of the model (equation 5). We have added this explanation to lines 330–331 in the revised manuscript.

*16.     Line 274. Should Ltide be a "tidal wave not under or free of the influence of Rwind"?*

To clarify what we mean, we have rewritten this line to "$L_{\text{tide}}$ is the distance travelled by the tidal wave along the shelf."

*17.     Line 285. Figure 3 should be described better. For instance, instead of putting Figure 3 in parenthesis, the sentence should start describing the figure. Figure 3 has 7 panels (a-g); why did the authors not mention them in the main text? This happens to most figures with several panels on them. There are no references to the different panels in the text. Please review all your figures and change them accordingly.*

We have rewritten this line: "The frequency distributions of $R_{\text{gauge}}$ at all seven locations are shown in Fig. 3, all of which deviate significantly from their respective No-TSI distribution…". We now also reference the panels/sub-figures of Fig. 3, 4, 5 directly whenever relevant.

**18.** *Line 290-298. From a physical point of view (e.g. local geographical set, physical processes), what could explain these differences in time of extreme residual occurrence? That is a good discussion point.*

Feng et al. (2019) suggested that this difference may be related to the ratio between the amplitude of tidal constituents $M_2$ and $K_1$, where extremes tend towards before HW at locations with larger $M_2$ and after HW when $M_2$ is small. We observe the same transition in our study region, where $M_2/K_1 < 1$ at the northernmost stations of Cendering and Geting, $M_2/K_1 = 1$ at Kuantan and $M_2/K_1 > 1$ at the remaining four stations to the south (Table 3). We have added this explanation to a paragraph describing spatial pattern in the TSI (lines 406–410).

**19.** *Line 299. The bootstrapping method should be explained better.*

The explanation of the bootstrapping method (Appendix A) has been improved.

*Why is it used?*

This method is chosen over the usual approach of the chi-square test due to our relatively small sample size $n$. As the chi-square test approximates the binomial distribution of $k_h$ with a normal distribution, a common rule of thumb requires $n \cdot p_h \geq 5$ for a decent approximation. This criterion is not satisfied for most values of $h$ at all seven locations. We have added this explanation to this line.

*Detailed information about it should be described in methods, not results.*

Explanation of the number of bootstrapping samples and family-wise error rate are rephrased and moved to the Methods section.

**20.** *Line 311. Which modes? What figure are you referring to? Are you talking about tidal components, like M2 and S2? Or are you talking about the bins at your histogram of frequency of occurrence? Where did you define modes previously in the text? Is that the statistical mode, e.g., the most frequent value of a distribution?*

We are referring to the statistical mode of the frequency distributions shown in Fig. 3–5. We have written "the primary mode of their frequency distributions" on the first mention of a mode (line 376).

**21.** *Line 316. Please reference the different panels in Figure 4.*

We have now referred to specific panels of Fig. 4 and others for better readability.

**22.** *Line 331-338. Please provide some possible physical explanations for the results.*

The shallow regional bathymetry and their proximity of tide-gauges to narrow waterways connecting the Pacific, Indian Ocean, and Java Sea are factors that enhance tide-surge interactions in the region. The results at two locations negative for tide-surge interactions are more likely due to a lack of sufficient data than an indication of the absence of any tide-surge interaction. We have added this explanation to the Results and Discussion section (lines 416–419, 423–424, 434–436).

**23.** *Line 340-343. This information repeats what was already said in the methods section. Please remove it from methods and keep it here.*

We have removed the repeated lines from the Methods section.

**24.** *Line 344-345. Are you comparing the timing of Rwind and No-TSI's timing (phase difference)? Or the distribution at which the extremes occur? It is not clear what you are comparing exactly. Please rewrite the sentence to make it clear.*

Here, we are doing the bootstrapping/hypothesis test on $R_{\text{wind}}$ to find if $R_{\text{wind}}$ contributes to TSI directly. These lines have been rewritten to make this clearer: "We obtain the timing of extreme $R_{\text{surge}}$ as a frequency distribution using the procedure described in Sect. 2.2 and compare it to its No-TSI distribution using hypothesis testing to determine whether there is any signal of tide-surge interaction in $R_{\text{surge}}$." Note that in our revised manuscript, we now account for the inverse barometer effect and hence we now have now replaced "$R_{\text{wind}}$" with "$R_{\text{surge}}$".

**25.** *Line 351. Figures S6-S9 need a better legend. Please replace "counts" with "Rwind" when applicable.*

The labels in the legends of the following figures have been changed: 3, 5, S3–S13 and S15–S17 (numbering based on the old manuscript). Note that we have made some changes to the presentation of the frequency distributions in the updated manuscript, so their legends will look different and contain a colour bar.

**26.** *Line 355. Figures S10-S13 need a better legend. Please replace "counts" with "Rphase" when applicable.*

The labels in the legends of the following figures have been changed: 3, 5, S3–S13 and S15–S17 (numbering based on the old manuscript).

**27.** *Line 366-373. Join this paragraph to the last one.*

We have combined the two paragraphs as suggested.

**28.** *Line 374. The authors are adding new methods to the results. This procedure should be added to methods, not here.*

We have moved this to the methods section (lines 356–359).

*Also, the purpose of Rsum is to compare with Rgauge. Please make it clearer.*

We made this clarification to the final sentence of the Methods section (lines 359–361).

**29.** *Line 378: "This can also be seen in Figure 4…". Actually, in Cendering, the median and mean are quite similar. Please correct it.*

Note that we have modified our semi-empirical model to account for the inverse barometer effect and the spatial variability of wind and bathymetry. Hence our results and discussion here have been amended slightly. Nonetheless, we clarify what we meant by our statement:

Our statement refers to Fig. 4b. In the boxplot for the frequency distribution of $R_{\text{sum}}$ at Cendering, the distribution median is $-2$ hours and the 95% CI of the median is the interval $[-3,0]$ hours (i.e. we cannot confidently conclude that the median is not zero). Based on this,

a visual comparison of the frequency distribution to the no-TSI distribution (Fig. S15f) and a statistical test (Fig. S15f), we conclude that there is no significant TSI at Cendering. We do not use the mean of the frequency distribution as it is not a very useful statistic for our analysis.

*30.*     *Line 380. In the sentence "We find that…", where is the result shown? In which figure?*

It is shown in Fig. 5. We now clarify this (line 483).

*31.*     *Line 381-394. This is a good discussion point. Why do diurnal tides not experience the TSI? Please develop a physical explanation for that.*

Thank you for the encouraging comment. Note that the change in methodology have slightly amended our results and discussions: for diurnal tidal cycles we find significant TSI at Cendering but not at Tioman, Kuantan and Geting. We provide the following explanations for our results:

- We lack sufficient observations at Tioman and Kuantan as diurnal tides are extremely rare and thus, we cannot confidently reject the null hypothesis.
- We do expect TSI during diurnal tides, which can be partially seen from our results for $R_{\text{phase}}$ during diurnal tides (Fig. S15 and S16). However, as the magnitude of $R_{\text{phase}}$ is proportional to the gradient of the tidal time-series, diurnal tides need to have twice the magnitude of semidiurnal tides (i.e. tidal form factor $\geq 2$) to have the same influence on $R_{\text{phase}}$. The tidal form factor of 1.9 at Cendering is close to 2, hence a stronger signal of TSI is present. Meanwhile, the tidal form factor of 1.56 at Geting is sufficiently lower than 2.

We explain this in detail in lines 423–424, 434–436 and 488–492 of the revised manuscript.

*32.*     *Line 389-394. I do not understand the connection between the last sentence and the rest of the paragraph. It is not clear what the point of discussion is. This last sentence should be the start of your paragraph (topic sentence), and your results (first two sentences of the paragraph) should support your idea. Please rewrite the paragraph to make your ideas and discussion clearer.*

This paragraph has been restructured based on this suggestion: "We find that $R_{\text{phase}}$ can significantly influence the distribution of the extreme values of $R_{\text{sum}}$, indicating that the process of tidal phase alteration—where surges perturb the depth of the water body and influences the propagation speed of the tide—produces a significant and measurable tide-surge interaction at five locations. This is despite $R_{\text{phase}}$ contributing to <1% of the variance of $R_{\text{sum}}$ at all seven tide gauge locations (Fig. S18). By an alternative metric, the ratio between the standard deviation of $R_{\text{phase}}$ and the standard deviation of $R_{\text{wind}}$ is only 0–1%. Thus, while the magnitude of $R_{\text{sum}}$ is effectively fully dependent on $R_{\text{wind}}$, the timing of its largest values is dependent on $R_{\text{phase}}$, indicating the significant contribution of tidal phase alteration to tide-surge interaction. Our findings complement those of Chen et al. (2012), who found that the influence of tide-surge interactions on large surges is negligible, although tide-surge interaction may alter the time of tidal high and low water."

*33.*     *Line 395. Please Remove "however".*

We have removed "however".

*34.     Line 395. Should I compare Rgauge and Rsum? If so, this needs to be better explained in the methods section and in the text.*

Yes, we now clarify this in the final sentence of the Methods section (see also comment 28).

*Which figures should I look at? Please refer to the text.*

We now reference Fig. 4 in the text.

*35.     Line 402-406. Interesting discussion point; however, the authors should develop it further. What could you change in your model to account for these limitations? Here, you should talk about why your model is limited, how you could improve it, and reference works that may have done it somehow. The authors cited 2 references but did not give much detail and explore it further.*

We now provide further discussion of the limitations of our model, how it can be improved, and relevant studies that used a hydrodynamical model. This discussion has been added to the revised manuscript in lines 512–550. (See also our response to main comments 1 & 5 on the Results and Discussion section, above.)

*I could see you used a constant depth to obtain Rwind. You should maybe account for shallower depths and several values. For instance, could you repeat the process by considering different depths? For example, you could calculate Lwind and Rwind from bathymetry 60m-40, then after 40-20, 20-10m... Do you think that would make a difference? Also, should we take different wind directions into account?*

We thank the reviewer for this helpful comment. We have since modified our semi-empirical model such that we no longer assume spatial homogeneity of wind and bathymetry depth i.e. we do the integration in Eq. (3). By doing so, we account for spatial variability in depth, wind speed and wind direction. We now also account for the inverse barometer effect in our semi-empirical model. We find that our final results and discussion remain qualitatively similar to before.

*Another point would be: Are there any local wind observations? You could use it instead of ERA5.*

We are interested in how wind forcing over the Sunda Shelf changes water depth over the shelf, thereby influencing the propagation of tides. For this large-scale wind forcing, a reanalysis dataset (such as ERA5) is more appropriate than local wind observations near the tide gauge.

*Maybe some conceptual figures could be placed here to help explain mutual phase alteration, etc....*

Due to the substantial number of figures and supplementary figures, we would like to avoid adding any more figures. As an alternative, we provide further elaboration on the process: "Changes in tides due to tidal phase alteration can affect the height of surges through Eq. (5), while surges further contribute to tidal phase alteration through Eq. (7) causing the mutually interacting process of mutual phase alteration between tides and surges."

*Another point is to discuss the other main mechanisms of tide surge interaction: advection, shallow water effects, and bottom friction. Are all your tide gauges located on the open coast? Or is any of them at a bay or harbour/estuary? That could have a massive impact on your results. Please see general comments on other important discussion points.*

The points here overlap with our response to main comment 1 on the Results and Discussion section (above). Further discussion can be found in lines 512–534 of the revised manuscript.

**36.    Line 408. I believe a similar approach was used by Arns et al. 2020.**

Our focus and approach differs from that of Arns et al. (2020). As we understand it, Arns et al. (2020) seek to quantify the contribution of TSI to the height of extreme of differences. In contrast, we seek to quantify the contribution of TSI to the timing of peak residuals.

**37.    Line 411:** *"No-TSI distribution should be used instead of a uniform distribution." I propose you perform a classification using a uniform distribution of your data and see how it performs compared to your approach. This would illustrate clearly what you want to show. It could be for only 1 or 2 sites. This would give you a good discussion point in the discussion section.*

We have re-run all hypothesis testing using a uniform distribution and chi-square tests for comparison. We have added the following findings to lines 540–542: "We found false positive signals for tide-surge interactions in $R_{\text{surge}}$ at Tanjong Pagar and Johor Baharu and in $R_{\text{sum}}$ at Geting when using the uniform distribution."

**38.    Line 423:** *"We found the residual component caused by advancement of tidal HW." Does the wind cause this? Please rewrite the sentence.*

Yes, it is caused by the wind which generates $R_{\text{surge}}$ (previously referred to as $R_{\text{wind}}$), which in turn generates $R_{\text{phase}}$. We have rewritten this sentence (lines 568–570): "we found that $R_{\text{phase}}$—the residual component caused by the advancement of tidal HW due to surges—can significantly alter the timing of extremes despite…"

**39.    Line 430-433.** *Please move this sentence to the discussion section. There are usually no citations in the conclusion section.*

The sentence has been moved to the Discussion section (lines 636-638).

**Referee #2**

**General comments**

*The authors investigate the tide-surge interaction near Singapore and Malaysia based on observation data and with a semi-empirical model.*

*They found tide-surge interaction at all studied locations based on observations. They conclude that the tidal phase could change but with minimum impact on high water. Their semi-empirical model could not simulate the tide-surge interaction.*

*The paper is well written but there are a lot of unclear points for me.*

Thank you for your comments, which have helped us to identify and address the unclear points, contributing to a clearer paper.

*1.    For whom, do they have written this paper?*
This paper is written for scientists, especially those interested in extreme sea level and tide-surge interaction.  By informing subsequent research, this paper may benefit practitioners who manage coastal infrastructure – but such practitioners are not our primary audience.

*2.    Why are the tide-surge interactions so important for this region if there is only a minimum change in water level. What does it mean e.g. for coastal protection or warning systems?*
In our revised manuscript, we affirm that "Understanding tidal dynamics, surge generation, and their mutual interaction is required to improve operational forecasts of sea levels and the statistical estimation of its extremes (Olbert et al., 2013; Horsburgh and Wilson, 2007; Tawn and Vassie, 1989)" (lines 44–46). Strong tide-surge interactions are expected in shallow regions such as the Sunda Shelf (lines 49–50). The expanse of the shallow shelf allows tidal phase alteration, one mechanism of tide-surge interaction, to develop before reaching the coast (lines 89–91). Improved understanding of this mechanism would improve water level prediction for operational forecast of sea level and aid development of statistical estimation techniques of sea-level extremes for coastal protection design purposes.  For example, the existence of TSI suggests that the joint probability method may not be suitable for characterising sea-level extremes, unless skew surge is used (lines 546-550).

*3.    There is no time-series of water level including the diurnal semi-diurnal tides from the different locations, e.g. for one or two weeks. The locations are so different in their water levels.*
We have added a figure to the revised supplementary materials (Fig. S3) to illustrate the time-series of the detrended water level, tide level and $R_{\text{gauge}}$ over the four days that contain the largest recorded value of $R_{\text{gauge}}$.

*4.    How good are the observation data?*
We use research quality data from the University of Hawaii Sea Level Center. We have clarified this in the revised manuscript (Lines 145, 149).

*5.    Is there a trend in the observation data?*
We have removed trends of annual scale or longer in our analysis by removing the 1-year moving average of the hourly water level. We have added the following (lines 168-169) to clarify this: "Detrending the hourly water level by the 1-year moving average removes trends or annual or longer timescales from the tide gauge records."

*6.    Are there changes in tidal range at the locations over the investigated period?*
We found small changes in the mean diurnal tidal range in the order of 0.2 cm at Tanjong Pagar to 4.9 cm at Geting. We have added this to lines 201-202 of the revised manuscript.

*7.    I'm wondering that data from a global database is used and no local data directly from the gauges (authorities) with e.g. 15-20 min resolution. Hourly observations could miss the low and high peaks.*

This is a good point and we thank the reviewer for bringing this up. We do not have access to higher resolution data along the coast of Malaysia. Additionally, we prefer to use publicly available data to support reproducibility and open science. Therefore, we employ the highest resolution data that is publicly available at these locations. Although this public data is at hourly resolution, Horsburg and Wilson (2007) showed that there is no qualitative difference between frequency distributions using 15-minute and hourly observations. Also, the use of hourly observations is consistent with other studies that have investigated TSI (e.g. Costa et al., 2023; Feng et al., 2019; Antony and Unnikrishnan, 2013; Haigh et al., 2010).

*8.      Do the authors test another tide model? How good is the tide model compared to the observation during calm wind situations?*
We did not test other tide models. We employ harmonic analysis as it is a robust and commonly-used method (Costa et al., 2023; Feng et al., 2019; Antony and Unnikrishnan, 2013; Olbert et al., 2013; Haigh et al., 2010; Horsburgh and Wilson, 2007). To this end, UTide is a popular package that is lightweight and robust in handling long time-series with irregular sampling rates (Opel et al., 2024; Costa et al., 2023; Torres and Nadal-Caraballo, 2021). As we use a long time-series (~15 years) when applying harmonic analysis, we believe UTide has provided us with a robust estimation of the tidal time-series.

*9.      How do the authors avoid making mistakes in the phase shift of the tides?*
To avoid such mistakes, we use research quality data from UHSLC and conduct harmonic analysis using long time-series (~15 years). Our method of harmonic analysis is consistent with other similar studies (Costa et al., 2023; Feng et al., 2019; Antony and Unnikrishnan, 2013; Olbert et al., 2013; Haigh et al., 2010; Horsburgh and Wilson, 2007). We also find that the inclusion of $R_{\text{phase}}$ improves the $R^2$ (coefficient of determination) of our semi-empirical model residuals compared with observations, indicating that phase shifts in our observed residuals are likely genuine signals of tide-surge interaction.

*10.     All results of the semi-empirical model are in the supplements. Why?*
The final results of the semi-empirical model are presented in Figs. 4b and 5. To keep the manuscript reader-friendly and concise, we choose to keep less important figures in the supplementary material. This is especially considering that most figures in the supplementary material take up almost one whole page. The five figures in the main text are the most important for illustrating our methodology and presenting our key results.

*11.     In general text and figures could be put more into context.*
We have added further discussion to provide more context, for example by indicating how our results can be useful in other regions of the world, and how they can inform future research and real-world applications (lines 535–550). This discussion includes the following points:

- Use of No-TSI distribution can improve hypothesis testing globally.
- The semi-empirical model could potentially be used at locations with similar local characteristics.
- Our results indicate that using skew surge may be more suitable than hourly residuals when estimating extremes of this region.

*12.     How high are the observed extreme water levels?*

In our revised manuscript, we include a brief description of surges in this region (lines 100–105). For example, we note that extremes of 0.35–1.0 m have been observed at Geting, the northernmost location of our study region.

*13.     And are they dangerous for the region?*

While the observed water levels do not necessarily pose a major hazard, understanding the underlying drivers is integral for applications such as early detection of extreme events, forecasting, or design purposes especially with sea level rise and the increasing frequency and intensity of storms (Mohd Anuar et al., 2018). We have included these points in the revised manuscript (lines 172–176).

*14.     Would you mind putting the bathymetry around the locations in table 1 or a higher resolution at the coast in figure 1. Is it steep or flat at the coastal locations?*

We have added this to the supplementary material as Fig. S1.

*15.     The influence of the bathymetry on surges is a little bit too short in the text.*

We have elaborated in the text that bathymetry affects surges through an inverse relationship from the denominator in Eq. (3): (lines 305–307): "Bathymetry also influences the height of surges through the term $D$ in the denominator of Eq. (3), causing shallow regions to experience larger surges when subject to the same wind forcing."

**More specific questions and comments:**

*16.     Are there regional hydrodynamic models available for this region to simulate surges and tide surge interaction, too? In line 72 to 79, you mentioned already some paper.*

Yes. In addition to the models that we have already mentioned, models that have been used for this region include the Singapore Regional Model (SRM) (Kurniawan et al., 2011), Delft3D-flow with an Alternating Direction Implicit finite difference scheme (van Maren and Gerritsen, 2012), and MIKE 21 (Mohd Anuar et al., 2023). Hydrodynamic modelling is an ongoing effort for this region. However, our goal here is to study tidal phase alteration, for which we have developed our semi-empirical model. Hence, we do not discuss any of the hydrodynamic models in detail in the main text.

*17.     Line 82: What kind of wave?*

We have removed this sentence now. We were initially referring to sea surface waves.

*18.     At the end of the Introduction a roadmap of the paper would be nice.*

Thank you for the suggestion. We have added a roadmap of our approach to the beginning of Sect. 2.

*19.     Line 131 to 135: Did you test other tide models?*

We did not test other tide models, as explained in our response to comment 8 above. To summarise, conducting tidal harmonic analysis using UTide is robust, appropriate for our study and consistent with other studies.

*20.     Line 157- 162: How long is a storm in this region? Can two extreme storms occur within seven days?*

Storm events can last between 1 and 5 days in this region (Meteorological Service Singapore; Tkalich et al., 2009). While there is a non-zero probability of two independent extreme events occurring within seven days, it is probably unlikely. We elaborate on the rationale behind our choice of a seven-day declustering period in the revised manuscript (lines 216-221).

*21.    Line 193 what are the differences between the specific locations? Unfortunately, I am not familiar with this region.*

We have now added the following details to help the readers:

- Maps showing the bathymetry in higher resolution (Fig. S1) and local characteristics (Fig. S2).
- A brief description of the specific locations (Sect. 2.1 lines 153–156): "Of our seven tide gauges, Tanjong Pagar and Johor Baharu are located within straits, Sedili at a river mouth, Tioman between Tioman island and mainland Peninsular Malaysia sheltered from the South China Sea, Kuantan at the mouth of a man-made bay, Cendering on the open coast and Geting sheltered inside a bay (Caldwell et al., 2001; OpenStreetMap contributors, 2017).

*22.    Line 201 -214 ???*

To make the derivation of the No-TSI distribution clearer, we have removed mathematical details that are potentially confusing and have edited qualitatively important details to be more reader-friendly (Section 2.3 & Appendix A).

*23.    Line 233-234 also for extreme water level? Is the wind direction always the same?*

Yes, this is also true for extreme water level. The prevailing wind direction in this region varies seasonally. We now mention these points in the same sentence (lines 303–305): "Monsoonal winds over the central South China Sea are the main determining factor of hourly non-tidal residuals and its extremes in the Singapore Strait (Tkalich et al., 2013a; Tkalich et al., 2009)." Our semi-empirical model accounts for variations in wind directions.

*24.    Line 237, what does it mean masking out the land? Are the locations now at different regions?*

When we used the term masking, it meant that we only include the 10m wind in our calculation of Eq. (4) if it is above water. If the data point is at a location above land, then we do not include it in our calculation. We now avoid using the terminology "masking" and have rewritten this line to "we use the region over water that is bounded by the red rectangle".

*25.    Line 243: How do you estimate the influence of bottom friction on the wind surge process with a depth of 58 m?*

We ignore the influence of bottom friction, an assumption that is reasonable except for very shallow waters below 30m (Pugh and Woodworth, 2014c). This condition is met at our study location except for a small region very close to the coastline. We highlight that our goal is not to produce accurate estimate of storm surges but to study the mechanistic effects of tidal phase alteration.

*26.    Line 262: The mean of wind speed over a large area of 759km x 833 km is probably different to wind speed with a spatial resolution of ~31km of ERA5. Please comment!*

Following reviewers' suggestions, we have modified our semi-empirical model such that we no longer assume spatial homogeneity of wind and bathymetry. We now account for spatial variability in depth, wind speed, and wind direction. We now also consider the inverse barometer effect.

Our earlier decision to assume spatial homogeneity (which turned out to be a reasonable assumption and produced similar model results) was that the spatial average can provide us the first-order estimate of storm surges, which will then allow us to produce and study a first-order estimate of the effects of tidal phase alteration.

*27. Why do the authors not use for every location their own wind speed and -direction in their semi-empirical model? ERA5 has a resolution of ~31km. I'm wondering that there is no hint about wind direction.*

We choose ERA5 data because we are interested in how wind forcing over the South China Sea changes water depth over the shelf, thereby influencing the propagation of tides. As we no longer assume spatial homogeneity, we have accounted for the spatial variability of both wind speed and wind direction in Eq. (3) and (4) through the term $\boldsymbol{W} \cdot \hat{\boldsymbol{x}}$.

*28. A storm tide is the sum of tides, wind surge, external surge, the inverse barometer effect, and non-linear tide–surge interactions. I miss the inverse barometer effect in your paper. (Please revise your definition in line 33-34).*

We have revised our definition to include the inverse barometer effect. This statement now reads: (lines 33–37) "Storm surges are rises in sea level brought about by low atmospheric pressure and strong winds acting on the sea surface (Pugh and Woodworth, 2014a). The destructive effects of large storm surges can be amplified by high tides if the surge coincides with high tide to produce large storm tides (Stephens et al., 2020; Gregory et al., 2019), increasing the risk of coastal flooding and threatening lives and properties (Pugh and Woodworth, 2014c)." We have now included the inverse barometer effect in our semi-empirical model.

**Results and Discussion**

*29. I am wondering that I have not seen any box plot of the water level and the residuals for every location.*

In our original manuscript, we did not discuss the height of water level or residuals. We have now included some discussion on their heights through the following:

- A new Table 2, which presents the standard deviation and certain key quantiles of the detrended water level.
- Revised frequency distributions, which now show the extreme residuals heights using coloured bar charts.
- A brief discussion on the height of residuals and how they impact storm tides (lines 366–373, 424–427).

In-depth discussion and analysis of the water levels is outside the scope of this study, which focus on the timing (not magnitude) of peak residuals and the associated mechanism of tidal phase alteration.

*30.    How big is the difference in water level between 99th and 95th (or 90th) percentile. Is the 95th percentile closer to the high-water events than the 99th percentile or is it the same. Are 90th percentile events more dangerous because, they can occur during high tide.*

Through the coloured bar charts, we now have some insights into the residual levels above the 99th percentile. We find a spatial pattern where extreme residuals are smaller at the Southern end of our study area and larger at the north. We also find that none of the largest residuals occurred near high water to form large storm tides (lines 366–373).

*31.    For Figure S2, I would prefer the same axis length and it should in the main part.*

Thank you for your suggestion. We have edited this figure for subplots to share axes length. We reiterate a reply given above (comment 10) that Fig. S2 serves to provide some validation to the semi-empirical model but does not add to the methodology, hence we put it in the supplementary materials.

*32.    S2 to S19 are in the supplements.*

We have provided our reason for doing this in our response to comment #10 above.

**Conclusions**

*33.    The authors provide a lot of statistics, but what do the results mean for decision-makers.*

Our paper is primarily intended for scientists who seek improved understanding of tide-surge interaction and extreme sea level. Although decision-makers are not our primary audience, improved understanding should of TSI should lead to improved knowledge of extreme sea level, supporting decision-making: "Knowledge of the interplay between tide-surge interaction and extreme sea levels can aid in the design of effective strategies for coastal planning, risk assessment, and mitigation measures, and will benefit from more comprehensive analyses" (lines 575–578).

*In summary, the paper needs some more investigations and clearer physical explanations.*

Informed by the reviewers' helpful comments, we have improved our methodology, conducted further investigations, provided clearer explanations of physical mechanisms, and have enhanced our discussions.

We thank both reviewers for providing detailed and constructive comments that have contributed to an improved manuscript.